# A point mutation in the nucleotide exchange factor eIF2B constitutively activates the integrated stress response by allosteric modulation

Morgane Boone[1,2†], Lan Wang[1,2†], Rosalie E Lawrence[1,2], Adam Frost[2,3], Peter Walter[1,2*], Michael Schoof[1,2*]

[1]Howard Hughes Medical Institute, University of California at San Francisco, San Francisco, United States; [2]Department of Biochemistry and Biophysics, University of California at San Francisco, San Francisco, United States; [3]Chan Zuckerberg Biohub, San Francisco, United States

**\*For correspondence:**
peter@walterlab.ucsf.edu (PW);
michael@walterlab.ucsf.edu (MS)

[†]These authors contributed equally to this work

**Abstract** In eukaryotic cells, stressors reprogram the cellular proteome by activating the integrated stress response (ISR). In its canonical form, stress-sensing kinases phosphorylate the eukaryotic translation initiation factor eIF2 (eIF2-P), which ultimately leads to reduced levels of ternary complex required for initiation of mRNA translation. Previously we showed that translational control is primarily exerted through a conformational switch in eIF2's nucleotide exchange factor, eIF2B, which shifts from its active A-State conformation to its inhibited I-State conformation upon eIF2-P binding, resulting in reduced nucleotide exchange on eIF2 (Schoof et al. 2021). Here, we show functionally and structurally how a single histidine to aspartate point mutation in eIF2B's β subunit (H160D) mimics the effects of eIF2-P binding by promoting an I-State like conformation, resulting in eIF2-P independent activation of the ISR. These findings corroborate our previously proposed A/I--State model of allosteric ISR regulation.

## Editor's evaluation

The paper describes the consequences of a missense mutation in the β subunit of the eIF2B complex that advances the understanding of the mechanisms of action of eIF2B in controlling the integrated stress response. The combination of biochemical, structural, and in-cell experiments constitutes a comprehensive study that supports a model for allosteric regulation of the active/inactive states of the eIF2B complex. The findings are relevant to neuropathologies, infectious and inflammatory diseases, diabetes, and metabolic disorders.

## Introduction

Coping with cellular stressors, manifesting as either intrinsic cues or environmental insults, is key to preserving cellular and organismal health. One strategy is to activate the integrated stress response (ISR), a conserved eukaryotic signaling network that reprograms translation toward damage mitigation and recovery, or apoptosis when stress is irremediable (*Costa-Mattioli and Walter, 2020*). The ISR integrates diverse stresses through at least four stress-sensing kinases – PERK, HRI, GCN2, PKR, and perhaps MARK2, via phosphorylation of a single serine, S51 of the α subunit of the translation initiation factor eIF2 (*Hinnebusch, 2005*; *Guo et al., 2020*; *Dey et al., 2005*; *Shi et al., 1998*; *Lu et al., 2021*). eIF2 is a central player in translation initiation, mediating start codon recognition on the

mRNA and delivery of the initiator methionine tRNA. Phosphorylation of eIF2 disrupts this process and leads to a precipitous drop in global protein synthesis. Conversely, the translation of a subset of stress-responsive mRNAs, such as *ATF4*, generally repressed by the presence of 5' UTR upstream open reading frames (uORFs), is induced (**Harding et al., 2000**). The alternative translation program, that is, thus set in motion drives the cell's return to homeostasis. While the ISR is inherently cytoprotective, its dysregulation has been documented in multiple disease states. Specifically, the ISR has been linked to neurodegenerative diseases (**Ma et al., 2013**), brain-injury induced dementia (**Chou et al., 2017**; **Sen et al., 2017**), aging (**Krukowski et al., 2020**), diabetes (**Abdulkarim et al., 2015**; **Harding et al., 2001**), and cancer (**Nguyen et al., 2018**; **Koromilas et al., 1992**).

Mechanistically, it is the level of ternary complex (TC) that determines the regulation of translation initiation by the ISR. The TC consists of eIF2 (heterotrimer composed of an α, β, and γ subunit, containing a GTPase domain in its γ subunit), the initiator tRNA loaded with methionine (Met-tRNA$^i$), and GTP (**Algire et al., 2005**). Once the TC associates with the 40 S ribosomal subunit, additional initiation factors, and the 5' methylguanine cap of the mRNA, the pre-initiation complex scans the mRNA for a start codon. Recognition of the start codon leads to GTP hydrolysis and triggers the release of eIF2 now bound to GDP (as reviewed in **Hinnebusch et al., 2016**). The large ribosomal subunit joins and the assembled 80 S ribosome proceeds with elongation of the polypeptide chain. Crucially, for every round of cap-dependent translation initiation, eIF2 requires GDP-to-GTP exchange, catalyzed by its dedicated guanine nucleotide exchange factor (GEF), eIF2B. Failure to complete this step impacts the cellular concentration of the TC, which impairs the translation of most mRNAs. At the same time, lower TC concentrations can induce the translation of specific stress-responsive ORFs, some of which are regulated by uORFs (**Harding et al., 2000**; **Lu et al., 2004**; **Vattem and Wek, 2004**). Thus, the ISR regulates translation by tuning the available pool of TC.

Given its central role in controlling TC levels and mRNA translation, many eIF2B mutations result in an aberrant ISR and severe disease, such as Vanishing White Matter Disease (VWMD) (**Leegwater et al., 2001**; **van der Knaap et al., 2002**). Molecularly, eIF2B is a large, heterodecameric complex composed of two copies each of an α, β, γ, δ, and ε subunit (**Kashiwagi et al., 2016**; **Tsai et al., 2018**; **Zyryanova et al., 2018**; **Wortham et al., 2014**; **Gordiyenko et al., 2014**). It has long been established that phosphorylation of eIF2 (eIF2-P) converts eIF2 from an eIF2B substrate to an eIF2B inhibitor, leading to a reduction in GEF activity and ISR activation (**Siekierka et al., 1982**; **Matts et al., 1983**; **Konieczny and Safer, 1983**; **Salimans et al., 1984**; **Rowlands et al., 1988**). Earlier atomic-resolution snapshots of the eIF2-bound and eIF2-P-bound human eIF2B complexes suggested steric hindrance to be the predominant mechanism for inhibition, given the proposed overlap of binding sites (**Kenner et al., 2019**; **Kashiwagi et al., 2019**; **Adomavicius et al., 2019**; **Gordiyenko et al., 2019**; **Bogorad et al., 2017**). However, we and others recently discovered that binding of the inhibitor eIF2-P to a distinct binding site — located on the face of the eIF2B complex opposite of the substrate-binding site — allosterically switches eIF2B from its active 'A-State' (which can readily engage eIF2 and catalyze nucleotide exchange) to an inhibited 'I-State' (**Schoof et al., 2021a**; **Zyryanova et al., 2021**).

The multi-subunit composition of eIF2B also lends itself to regulation at the level of complex assembly. The decameric holoenzyme is built from two eIF2Bβγδε tetramers and one eIF2Bα$_2$ dimer (**Tsai et al., 2018**). The eIF2Bε subunit harbors the enzyme's catalytic center but only contains a small part of the binding surface of eIF2. Two of four interfaces between eIF2 and eIF2B (IF1 and IF2) reside in eIF2Bε. Thus, poor substrate-binding severely limits eIF2Bε's catalytic activity. The substrate-binding surface is increased upon addition of more subunits (a third interface, IF3 in eIF2Bβ). Yet, even when embedded in the eIF2Bβγδε tetramer subcomplex, the specific enzyme activity ($k_{cat}/K_M$) of eIF2Bε is ~100 fold lower compared to the fully assembled eIF2B(αβγδε)$_2$ decamer (tetramer $k_{cat}/K_M$ = 0.07 min$^{-1}$μM$^{-1}$, decamer $k_{cat}/K_M$ = 7.24 min$^{-1}$μM$^{-1}$), in which the substrate-interacting surface is further extended by bridging the twofold symmetric interface formed between the two tetrameric subcomplexes (a fourth interface, IF4 in eIF2Bδ') (**Schoof et al., 2021a**; **Kenner et al., 2019**; **Kashiwagi et al., 2019**). eIF2B activity, assembly-state, and conformation are all modulated by the ISR inhibitor, ISRIB. This small molecule binds in a deep groove spanning across the symmetry interface of the two eIF2B tetramers and enhances its GEF activity (**Sekine et al., 2015**; **Sidrauski et al., 2013**; **Sidrauski et al., 2015**; **Tsai et al., 2018**; **Zyryanova et al., 2018**). ISRIB exerts these effects by acting on both eIF2B assembly and conformation (**Schoof et al., 2021a**). When eIF2Bα$_2$ levels are low, pharmacological dimerization of tetrameric subcomplexes by ISRIB rescues eIF2B function (**Schoof et al., 2021a**).

When eIF2B$\alpha_2$ levels are saturating and eIF2B decamers are therefore fully assembled, ISRIB binding stabilizes eIF2B in the active 'A-State', reducing its affinity for the inhibitor eIF2-P (*Schoof et al., 2021a*; *Zyryanova et al., 2021*).

Given these insights, we here revisit previous observations concerning a histidine to aspartate point mutation in eIF2Bβ (βH160D) that straddles the junction of the β-β'and β-δ' interface (the ' notation indicates that the subunit resides in the adjoining, second tetramer in eIF2B) (*Tsai et al., 2018*). We formerly observed that this missense mutation blocked ISRIB-driven assembly of eIF2B tetramers into octamers in vitro, underlining the importance of the H160 residue in stabilizing the octamer (*Tsai et al., 2018*). However, whether the change to aspartic acid, predicted to be repulsed by the apposed D450 in δ', precluded decameric assembly or activated the ISR, remained unknown. Here, we show that the βH160D mutation does not affect decameric holoenzyme formation when all subunits are present. However, this mutation stabilizes eIF2B in an inactive conformation reminiscent of the inhibited 'I-State', normally promoted by eIF2-P binding. Concomitantly, cells with this mutation constitutively activate the ISR, even in absence of stress and eIF2-P. These results validate the A/I-State model of eIF2B and ISR regulation by showing that a conformational change in eIF2B is sufficient to impair its enzymatic function and activate the ISR.

## Results
### The eIF2B βH160D mutation does not block decamer assembly in vitro

To dissect the regulation of eIF2B assembly and activity, we purified human eIF2Bβδγε tetramers both with and without the βH160D mutation (*Figure 1—figure supplement 1*). We first performed sedimentation velocity experiments to assess the assembly state of eIF2B. Consistent with our previous

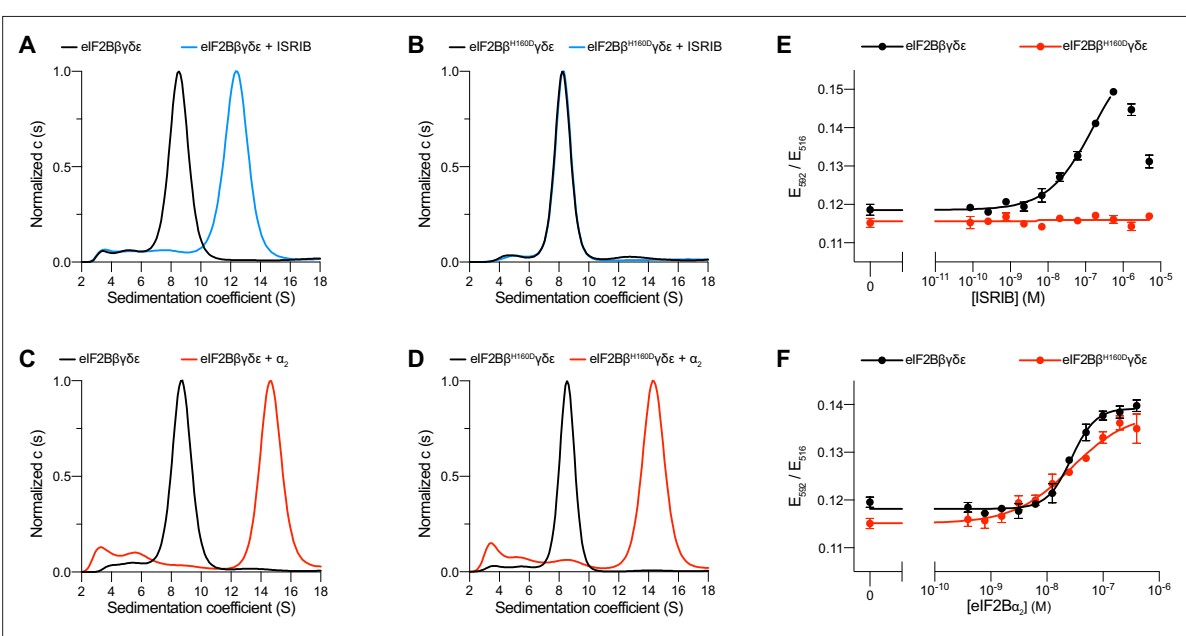

**Figure 1.** The eIF2B βH160D mutation prevents octamer assembly but not decamer assembly. (**A–D**) Characterization by analytical ultracentrifugation (sedimentation velocity) of (**A**) 500 nM eIF2Bβδγε ± 1 μM ISRIB, (**B**) 500 nM eIF2Bβ$^{H160D}$δγε ± 1 μM ISRIB, (**C**) 500 nM eIF2Bβδγε ± 500 nM eIF2Bα$_2$, and (**D**) 500 nM eIF2Bβ$^{H160D}$δγε ± 500 nM eIF2Bα$_2$. The eIF2Bβδγε tetramer sediments with a sedimentation coefficient of ~8 S, the eIF2B(βδγε)$_2$ octamer at ~12 S, and the eIF2B(αβγε)$_2$ decamer at ~14 S. (**E–F**) FRET signal (E$_{592}$/E$_{516}$) measured after 1 hr of eIF2Bβδγε-F tetramers incubation with (**E**) ISRIB or (**F**) eIF2Bα$_2$. For assembly by ISRIB, WT EC$_{50}$ = 170 ± 25 nM. For assembly by eIF2Bα$_2$, WT EC$_{50}$ = 29 ± 3 nM and βH160D EC$_{50}$ = 33 ± 3 nM. WT and βH160D eIF2Bβδγε-F tetramers at 50 nM throughout. For (**E,F**), representative replicate averaging four technical replicates are shown. Biological replicates: n = 3. All error bars and '±' designations are s.e.m.

The online version of this article includes the following source data and figure supplement(s) for figure 1:

**Source data 1.** Raw data for AUC and FRET experiments.

**Figure supplement 1.** Coomassie-stained gel of purified proteins used in this study.

**Figure supplement 1—source data 1.** Original image file for SDS-PAGE gel.

observations (*Tsai et al., 2018*), WT eIF2B tetramers readily assembled into octamers in the presence of ISRIB, whereas βH160D tetramers did not (*Figure 1A, B*). In contrast, we found that assembly into the fully decameric holoenzyme by addition of the eIF2Bα$_2$ dimer was not compromised (*Figure 1C, D*).

To confirm that the βH160D mutation does not impair decamer assembly, we utilized an orthogonal, previously established Förster resonance energy transfer (FRET) assay to assess eIF2B's assembly state (*Schoof et al., 2021a*). In this system, the C-terminus of eIF2Bβ is tagged with mNeonGreen as the FRET donor and the C-terminus of eIF2Bδ with mScarlet-i as the FRET acceptor. Both WT and βH160D tetramers were purified with these fluorescent tags (and hereafter are denoted eIF2Bβδγε-F). A titration of ISRIB readily assembled WT eIF2Bβδγε-F tetramers into octamers (EC$_{50}$ = 170 ± 25 nM) but did not promote βH160D eIF2Bβδγε-F assembly into octamers, even at the highest concentrations tested (*Figure 1E*). By contrast and in agreement with the analytical ultracentrifugation data in *Figure 1A–D*, titration of eIF2Bα$_2$ assembled both WT (EC$_{50}$ = 29 ± 3 nM) and βH160D (EC$_{50}$ = 33 ± 3 nM) eIF2Bβδγε-F tetramers into decamers with comparable efficiency (*Figure 1F*).

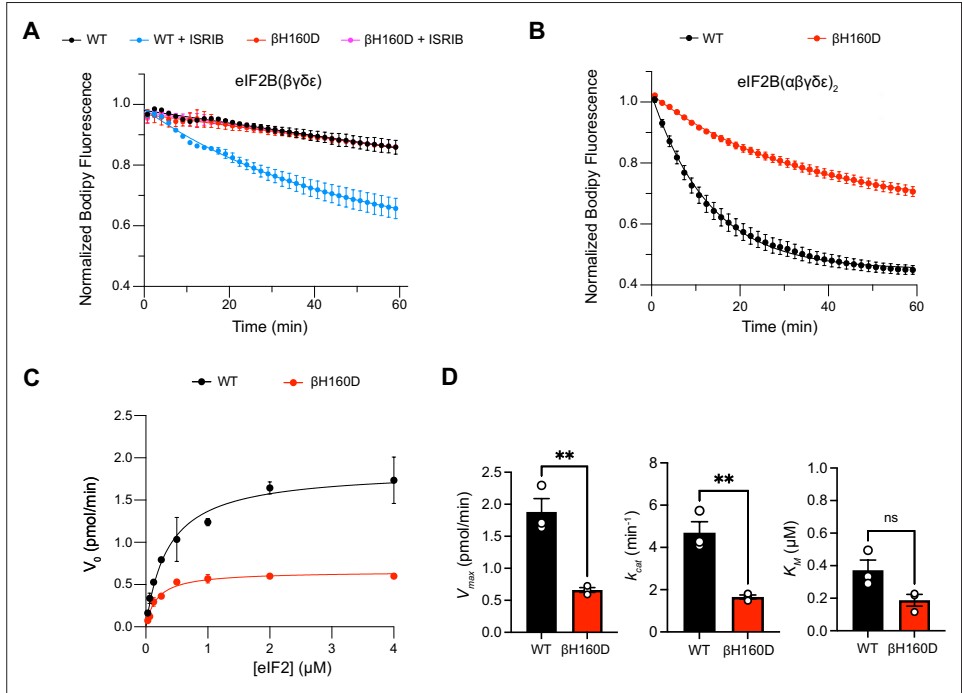

**Figure 2.** The βH160D mutation impairs nucleotide exchange by the eIF2B holoenzyme. (**A,B**) GEF activity of eIF2B as assessed by BODIPY-FL-GDP exchange on eIF2 using (**A**) eIF2B tetramer (100 nM) and (**B**) eIF2B decamer (10 nM). For (**A**), ISRIB only stimulates eIF2B guanine nucleotide exchange (GEF) activity for the WT tetramer (t$_{1/2}$ = 31.1±1.47 min). In (**B**), the βH160D decamer has lower GEF activity (t$_{1/2}$ = 23.57 ± 0.82 min) than WT decamer (t$_{1/2}$ = 9.28 ± 0.96 min). (**C**) Michaelis–Menten fit of the initial velocity of eIF2B-catalyzed nucleotide exchange as a function of eIF2 concentration (10 nM eIF2B decamer throughout). (**D**) Kinetic parameters of the Michaelis–Menten fit. βH160D decamers have ~threefold reduced intrinsic enzymatic activity (WT $V_{max}$ = 1.86 ± 0.13 pmol min$^{-1}$; βH160D $V_{max}$ = 0.66 ± 0.03 pmol min$^{-1}$; two-sided $t$-test p=0.0045) and turnover number (WT $k_{cat}$ = 4.70 ± 0.52 min$^{-1}$; βH160D $k_{cat}$ = 1.65 ± 0.10 min$^{-1}$; two-sided $t$-test p=0.0045). The $K_M$ is not significantly different (WT $K_M$ = 0.36 μM ± 0.09 μM; βH160D $K_M$ = 0.18 ± 0.03 μM; two-sided $t$-test p=0.07). Biological replicates: $n$ = 2 for (**A**), and $n$ = 3 for (**B–D**). All error bars and '±' designations are s.e.m.

The online version of this article includes the following source data and figure supplement(s) for figure 2:

**Source data 1.** Raw data for nucleotide exchange assays.

**Figure supplement 1.** The βH160D mutation decreases the initial velocity of eIF2B's guanine nucleotide exchange factor (GEF) activity.

**Figure supplement 1—source data 1.** Raw data for Michaelis–Menten analysis of nucleotide exchange at various eIF2 concentrations.

## The eIF2B βH160D decamer is impaired in GEF Activity

These properties are reminiscent of eIF2B's behavior in the presence of its inhibitor eIF2-P. In the inhibited decameric conformation (I-State) induced by eIF2-P binding, ISRIB binding to eIF2B is impaired (*Schoof et al., 2021a*; *Zyryanova et al., 2021*). We next asked whether the βH160D mutation impacts eIF2B's enzymatic activity. To this end, we monitored eIF2B's GEF activity using a BODIPY-FL-GDP nucleotide exchange assay. Both WT and βH160D tetramers exhibited comparably low enzymatic activity. The activity was robustly enhanced in WT octamers assembled from tetramers with ISRIB but, as expected, ISRIB had no impact on βH160D tetramer activity (*Figure 2A*). Intriguingly, βH160D decamers were less active than WT decamers ($t_{1/2}$ = 23.6 ± 0.8 min vs. 9.3 ± 1.0 min, respectively) (*Figure 2B*). To understand how the βH160D mutation impaired eIF2B's GEF activity, we next performed nucleotide exchange assays of WT and βH160D decamer activity at varying eIF2 concentrations. We measured the initial velocity of these reactions and fit the data to the Michaelis-Menten model of enzyme kinetics to determine the $V_{max}$ and the $K_M$ of the nucleotide loading reaction (*Figure 2C*, *Figure 2—figure supplement 1*). The $V_{max}$ (and consequently also the $k_{cat}$) was significantly diminished by ~three fold for βH160D decamers when compared to WT decamers (WT $V_{max}$ = 1.86 ± 0.13 pmol min$^{-1}$; βH160D $V_{max}$ = 0.66 ± 0.03 pmol min$^{-1}$, two-sided *t*-test p=0.0045) suggesting that the βH160D mutation limits the intrinsic enzymatic activity of eIF2B (*Figure 2D*). In contrast, we could not detect a significant difference in measured $K_M$ (WT $K_M$ = 0.36 ± 0.06 μM, βH160D $K_M$ = 0.19 ± 0.04 μM, two-sided *t*-test p=0.07).

## Impaired substrate binding in decameric eIF2B results from the βH160D Mutation

The absence of a clear difference in $K_M$ was puzzling, as we suspected the βH160D decamer to adopt an inhibited conformation reminiscent of the I-State, where both intrinsic enzymatic activity and binding of eIF2 are compromised (*Schoof et al., 2021a*). We therefore directly assessed binding affinities of eIF2B's substrate (eIF2) and inhibitor (eIF2-P), using surface plasmon resonance (SPR) to measure binding to WT decamers, βH160D decamers, and WT tetramers. eIF2 association with WT and βH160D decamers was monophasic, but dissociation was notably biphasic irrespective of eIF2 concentration, with a fast phase and a slow phase (*Figure 3A, B*). Although the rate constants $k_a$, $k_{d\,fast}$, and $k_{d\,slow}$ were broadly comparable, eIF2 binding to WT vs. βH160D decamers differed in the percentage of fast phase dissociation events (WT = 29%; βH160D = 67%) (*Figure 3A, B*, *Table 1*). Thus, a larger fraction of substrate molecules dissociates rapidly from βH160D compared to WT decamers. Since the $K_M$ is only equal to the $K_D$ when the dissociation rate constant $k_d$ is much larger than the $k_{cat}$, this measurement can resolve the paradox of a similar $K_M$ but different dissociation behavior.

In contrast to eIF2's interaction with decameric eIF2B, binding to WT tetramers could be modeled using one phase association and dissociation. Indeed, eIF2 dissociation from tetrameric eIF2B can be thought of as being 100% fast phase as the dissociation constant was indistinguishable from the fast phase dissociation constant for both WT and βH160D ($k_d$ = 0.12 s$^{-1}$) (*Figure 3C*). The fraction of eIF2 molecules that dissociate from decamers with fast phase kinetics may therefore only be engaging eIF2B through interfaces 1–3 (interfaces 1 and 2 in eIF2Bε and interface 3 in eIF2Bβ). In contrast, the eIF2 molecules that dissociate with slow phase kinetics may additionally contact interface 4 in eIF2Bδ', reaching across the central symmetry interface (*Schoof et al., 2021a*). This explanation would be consistent with identical dissociation constants for tetramer dissociation and fast phase dissociation from the decamers. For eIF2-binding, the βH160D decamers can therefore be thought of as more like isolated tetramers. That is, eIF2 readily associates but then is likely to dissociate too rapidly for efficient catalysis.

We further interrogated the biphasic dissociation behavior of eIF2 from WT and βH160D decamers by varying the time allowed for eIF2 binding to eIF2B (*Figure 3—figure supplement 1A, B*). For both WT and βH160D we observed an exponential decrease in the percentage of fast phase dissociation, which within two minutes plateaued at ~11% fast phase dissociation for eIF2 binding to WT and at ~55% fast phase dissociation for eIF2 binding to βH160D decamers (*Figure 3G*). These data argue that at equilibrium the fast phase dissociation plays a small part in the engagement between eIF2 and WT eIF2B but plays a significant part in substrate engagement with βH160D decamers. This kinetic behavior can be explained by a model proposing stepwise engagement between eIF2 and eIF2B in a process that first entails engagement of 3 interaction interfaces (IF1-3), followed by a second, slower

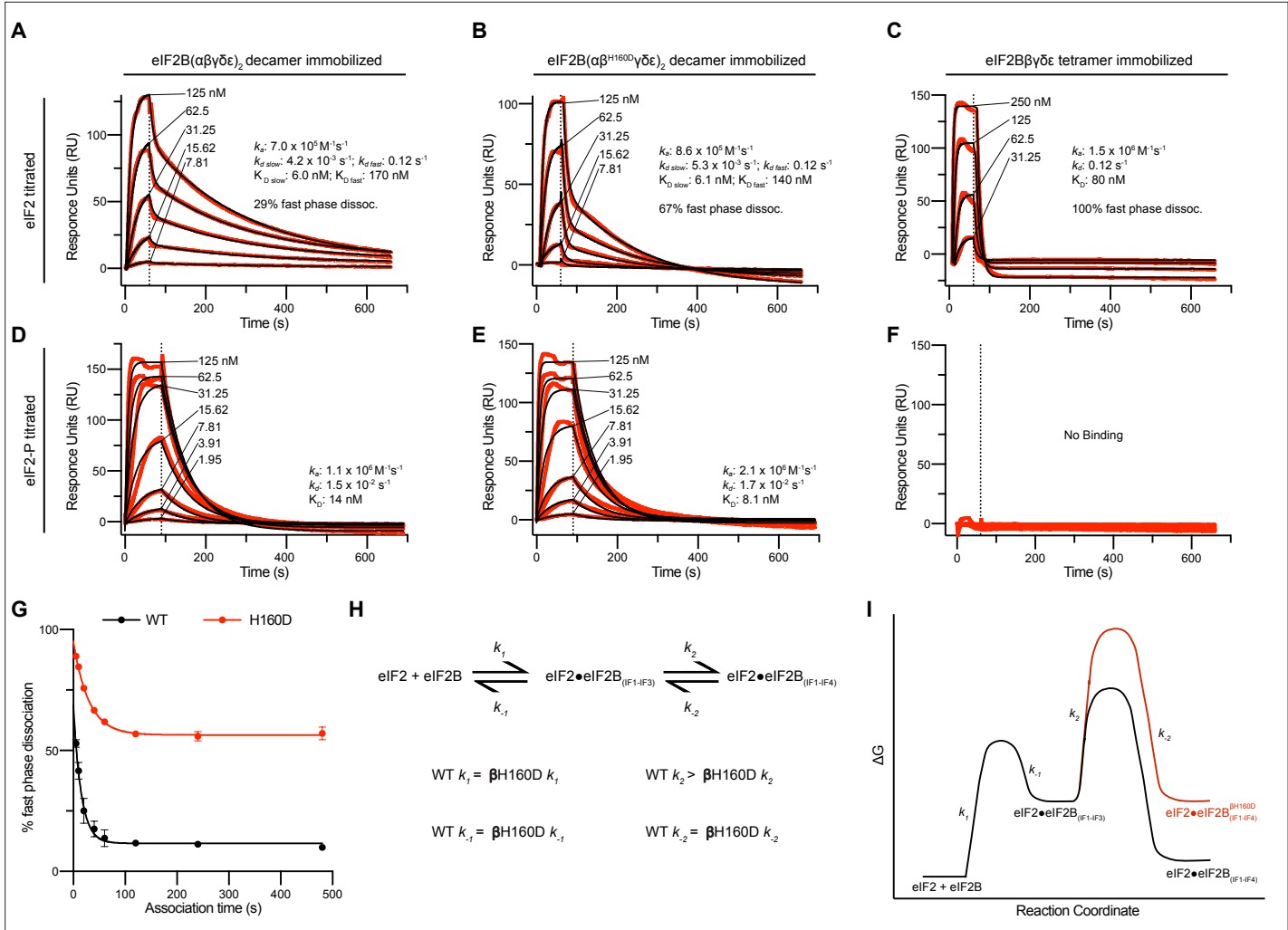

**Figure 3.** Substrate (eIF2) binding to eIF2B is compromised by the βH160D mutation. (**A–F**) Surface plasmon resonance (SPR) of immobilized (**A and D**) WT eIF2B decamer, (**B and E**) βH160D eIF2B decamer, and (**C and F**) WT eIF2B tetramer binding to 2-fold titrations of (**A–C**) eIF2 or (**D–F**) eIF2-P. For WT eIF2B decamer and βH160D eIF2B decamer, eIF2Bα was Avi-tagged and biotinylated. For WT eIF2B tetramer, eIF2Bβ was Avi-tagged and biotinylated. Binding was modeled as one-phase association for (**A–E**), two-phase dissociation for (**A–B**), and one-phase dissociation for (**C–E**). (**G**) SPR of immobilized WT eIF2B decamer and βH160D eIF2B decamer was performed with eIF2 at 62.5 nM throughout and varied association time from 5 to 480 s. The dissociation kinetics were then modeled (individual traces shown in *Figure 3—figure supplement 1*) and from this data percent fast phase dissociation was plotted as a function of association time with a single exponential fit. WT $t_{1/2}$ = 10.4 s; βH160D $t_{1/2}$ = 20.7 s. Percent fast phase dissociation is always higher for βH160D decamers vs. WT decamers and reaches an equilibrium at ~55% fast phase dissociation for βH160D decamers and ~11% fast phase dissociation for WT decamers. (**H**) Model reaction scheme of eIF2 engagement with eIF2B. $k_1$, $k_{-1}$, and $k_{-2}$ each are comparable for WT and βH160D decamers but WT $k_2$ > βH160D $k_2$. Based on the SPR data in *Figure 3A–C*, $k_1$ ~ 7.0 x $10^5$ $M^{-1}s^{-1}$ and $k_{-1}$ ~0.12 $s^{-1}$. $k_{-2}$ is calculated under the assumption that slow phase dissociation represents the combination of $k_{-1}$ and $k_{-2}$ dissociation. $k_{-1}$ is fast phase dissociation, so $k_{-1}$ = $kd_{d\,fast}$. Hence from $k_{-1} * k_{-2} = kd_{d\,slow}$ we get that 0.12 $s^{-1} * k_{-2}$ = 5.3 x $10^{-3}$ $s^{-1}$. Therefore $k_{-2}$ ~0.044 $s^{-1}$. (**I**) Free energy profile of eIF2 engagement with eIF2B either in the WT (black) or βH160D (black then red) context. Initial 3 interface engagement is energetically the same for either WT or βH160D, but engagement with the 4th interface is disfavored in the βH160D mutant. The free energy profile is drawn at sub saturating conditions. Given the percent fast phase vs slow phase dissociation at equilibrium in *Figure 1G* we know that for WT, [eIF2•eIF2B$_{(IF1-IF4)}$]/[eIF2•eIF2B$_{(IF1-IF3)}$] ~ 8 while for βH160D [eIF2•eIF2B$_{(IF1-IF4)}$]/[eIF2•eIF2B$_{(IF1-IF3)}$] ~ 1. For (**G**), n = 3 biological replicates. All error bars and '±' designations are s.e.m.

The online version of this article includes the following source data and figure supplement(s) for figure 3:

**Source data 1.** Raw data for SPR assays.

**Figure supplement 1.** The βH160D mutation increases the fraction of eIF2 molecules that bind and then dissociate with fast phase kinetics.

**Figure supplement 1—source data 1.** Raw data for eIF2 binding assessed by SPR using varying association times.

**Table 1.** SPR measurements of affinity.

| | eIF2 binding | | | eIF2-P binding | | |
|---|---|---|---|---|---|---|
| | **WT decamer** | **βH160D decamer** | **WT tetramer** | **WT decamer** | **βH160D decamer** | **WT tetramer** |
| $k_a$ (M$^{-1}$s$^{-1}$) | $7.0 \times 10^5$ | $8.6 \times 10^5$ | $1.5 \times 10^6$ | $1.1 \times 10^6$ | $2.1 \times 10^6$ | No binding |
| $k_d$ (s$^{-1}$) | Slow: $4.2 \times 10^{-3}$ Fast: 0.12 | Slow: $5.3 \times 10^{-3}$ Fast: 0.12 | 0.12 | $1.5 \times 10^{-2}$ | $1.7 \times 10^{-2}$ | No binding |
| $K_D$ (nM) | Slow: 6.0 Fast: 170 | Slow: 6.1 fast: 140 | 80 | 14 | 8.1 | No binding |
| % Slow dissociation | 71 | 33 | 0 | NA | NA | No binding |
| % Fast dissociation | 29 | 67 | 100 | NA | NA | No binding |

step that engages the fourth interaction interface (IF4; *Figure 3H, I*). In this model, the βH160D mutation does not affect the on/off rates of eIF2 engagement with eIF2B through interfaces 1–3, but slows the on-rate ($k_2$ in *Figure 3H, I*) of converting from 3 interface engagement to four interface engagement. Such a mechanism can explain the accumulation of the 'intermediate' fast phase dissociation species.

We next assessed eIF2-P binding to the immobilized eIF2B species. For both WT and βH160D decamer binding, this interaction could be modeled using one-phase association and dissociation kinetics. The overall affinity of eIF2-P for both species was largely comparable (WT $K_D$ = 14 nM; βH160D $K_D$ = 8.1 nM) (*Figure 3D, E*). As expected owing to the absence of the dimeric eIF2Bα subunit, which constitutes part of the eIF2-P binding site, we observed no noticeable eIF2-P binding to WT tetramers (*Figure 3F*).

From these results, we conclude that the βH160D decamer shares a number of properties with the eIF2-P-bound decamer: (1) reduced intrinsic GEF activity, (2) impaired substrate binding, and (3) insensitivity to ISRIB. Owing to these similarities, we wondered whether the βH160D mutation mimics eIF2-P binding and shifts eIF2B into an I-State or 'I-State like' conformation. To assess this notion, we determined the structure of the βH160D eIF2B decamer using single-particle cryo-EM.

## The βH160D mutation shifts eIF2B into an inhibited conformation

We prepared the βH160D decamer by combining βH160D tetramers and eIF2Bα$_2$, and subjected the sample to cryo-EM imaging. After 2D and 3D classification, we generated a single consensus structure of the βH160D decamer at 2.8 Å resolution (*Table 2*, *Figure 4—figure supplement 1*) with most side chains clearly resolved (*Figure 4A*, *Figure 4—figure supplement 1E, F*). This map allowed us to build an atomic model of how the βH160D substitution alters the conformation of the eIF2B decamer. By superimposing the βH160D decamer structure and our previously published A-State structure (eIF2B-eIF2 complex, PDB ID: 6O81), we observed a significant difference in their overall architecture: the two tetramer halves of the βH160D decamer underwent a rocking motion that changed the angle between them by approximately 3.5° (*Figure 4B*). This rocking motion repositions the two tetramer halves in an orientation comparable to the I-State structure (eIF2B-eIF2αP complex, PDB ID: 6O9Z), although not reaching the 6° angle observed for the eIF2-P-inhibited decamer (*Figure 4—figure supplement 2*). To further understand how the βH160D mutation affects the conformation and dynamics of the decamer, we performed additional cryo-EM analysis of both the WT and the βH160D decamer particles (*Figure 4—figure supplements 3–5*). We found the following: (1) in both the WT and the mutant, the two tetrameric halves can undergo rocking motions around the central axis; (2) the βH160D mutation shifts the mean conformation of the decamer towards the I-State; and (3) the βH160D dataset likely represents particles that follow a continuous conformation distribution, rather than a mixture of distinct A and I-State populations. These observations validate our hypothesis that the βH160D mutation shifts eIF2B from the active conformation towards an inhibited conformation.

We next examined changes to the ISRIB-binding pocket. Comparing the βH160D decamer to A-State (eIF2-bound eIF2B) and I-State (eIF2α-P-bound eIF2B) structures, we noticed that its ISRIB binding pocket was 3.3 Å wider in its long dimension than that of the A-State (*Figure 5A*), again reminiscent of the I-State (*Figure 5C*). The widening of the binding pocket can explain why ISRIB fails to assemble βH160D tetramers into octamers or affect GEF activity.

**Table 2.** Cryo-electron microscopy dataset for eIF2B$^{\beta H160D}$ decamer.

| Structure | eIF2B$^{\beta H160D}$ (PDB ID: 7TRJ) |
| --- | --- |
| **Data collection** | |
| Microscope | Titan Krios |
| Voltage (keV) | 300 |
| Nominal magnification | 105,000x |
| Exposure navigation | Image shift |
| Electron dose (e$^-$Å$^{-2}$) | 67 |
| Dose rate (e$^-$/pixel/sec) | 8 |
| Detector | K3 summit |
| Pixel size (Å) | 0.835 |
| Defocus range (μm) | 0.6–2.0 |
| Micrographs | 2,269 |
| **Reconstruction** | |
| Total extracted particles (no.) | 1,419,483 |
| Final particles (no.) | 170,244 |
| Symmetry imposed | C1 |
| FSC average resolution, masked (Å) | 2.8 |
| FSC average resolution, unmasked (Å) | 3.8 |
| Applied B-factor (Å) | 81.7 |
| Reconstruction package | Cryosparc 2.15 |
| **Refinement** | |
| Protein residues | 3,234 |
| Ligands | 0 |
| RMSD Bond lengths (Å) | 0.003 |
| RMSD Bond angles (°) | 0.838 |
| Ramachandran outliers (%) | 0.13 |
| Ramachandran allowed (%) | 3.62 |
| Ramachandran favored (%) | 96.25 |
| Poor rotamers (%) | 6.92 |
| CaBLAM outliers (%) | 2.50 |
| Molprobity score | 2.40 |
| Clash score (all atoms) | 9.59 |
| B-factors (protein) | 100.54 |
| B-factors (ligands) | N/A |
| EMRinger Score | 2.52 |
| Refinement package | Phenix 1.17.1-3660-000 |

Zooming in on the tetramer-tetramer interface, we examined the interactions in the WT eIF2B A-State decamer that stabilize the dimerization interface (*Figure 5B*). In the WT decamer, βH160 forms a π–π stacking interaction with δ'F452, which is lost in the βH160D eIF2B decamer and leads to the retraction of the short loop bearing this residue (*Figure 5B* and *Figure 5—figure supplement 1*). Other interactions in WT decamer include an ionic interaction between β'R228 and δ'D450, as well as a cation-π interaction between β'R228 andδ'F452. In the βH160D decamer, β'R228 repositions itself within the network of three negative charges (βE163, βD160, and δ'D450) and one aromatic amino

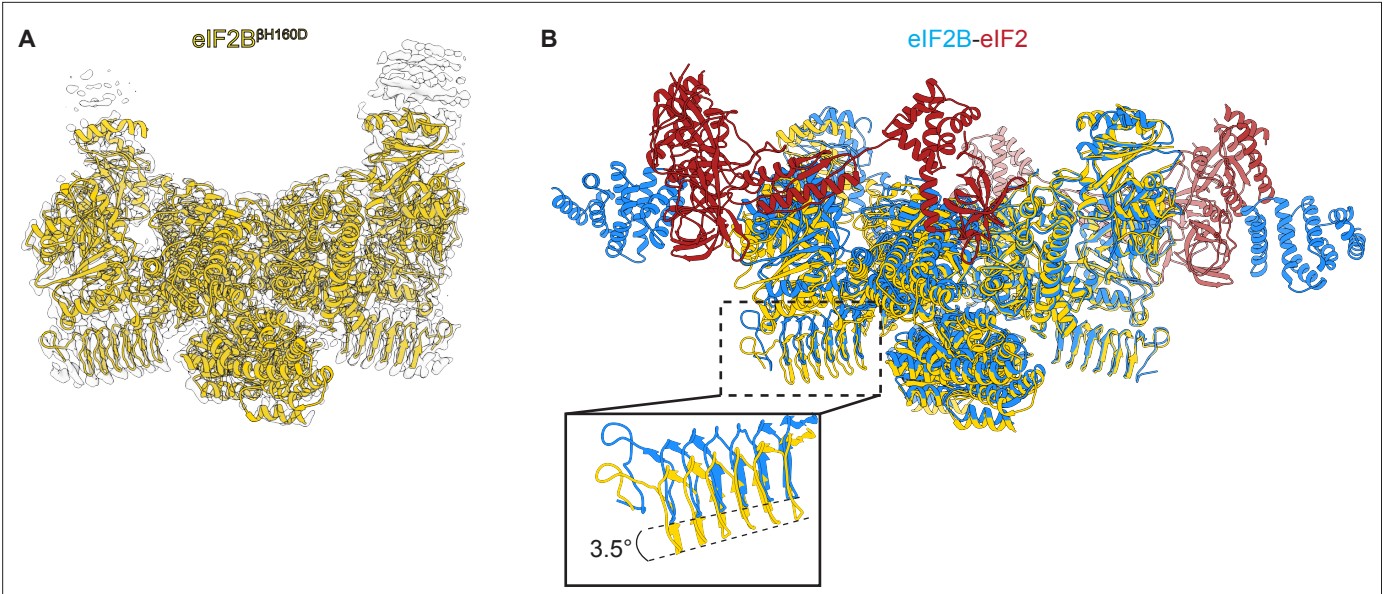

**Figure 4.** Overall architecture of eIF2B$^{\beta H160D}$. (**A**) Atomic model of eIF2B$^{\beta H160D}$ decamer (yellow) superimposed into the cryo-EM map (grey), showing the overall structure of the molecule. (**B**) Overlay of the eIF2B$^{\beta H160D}$ structure to the eIF2B-eIF2 structure (PDB ID: 6O81) shows a 3.5° hinge movement between the two eIF2B halves. eIF2B$^{\beta H160D}$ is shown in gold; eIF2B in the eIF2B-eIF2 structure in blue; eIF2 in red.

The online version of this article includes the following figure supplement(s) for figure 4:

**Figure supplement 1.** Cryo-EM data analysis of the eIF2B$^{\beta H160D}$ structure.

**Figure supplement 2.** Structure overlay of the A and I state models.

**Figure supplement 3.** Cryo-EM analysis of the conformation and dynamics of the WT decamer and the βH160D decamer – part 1.

**Figure supplement 4.** Cryo-EM analysis of the conformation and dynamics of the WT decamer and the βH160D decamer – part 2.

**Figure supplement 5.** Cryo-EM analysis of the conformation and dynamics of the WT decamer and the βH160D decamer – part 3.

acid (δ'F452) to reach a new stable state locally. The loop movement caused by the mutation propagates across the entire tetramer, resulting in the rocking motion observed in *Figure 4B*. This explains how the βH160D amino acid change in eIF2B remodels the dimerization interface to widen the ISRIB binding pocket and induce an I-State like conformation.

To further examine the long-range effect of this interface mutation, we looked at the critical interfaces for substrate (eIF2) binding provided by eIF2Bβ and eIF2Bδ. An overlay of the βH160D decamer structure with the eIF2B-eIF2 complex structure revealed that the substrate eIF2α binding pocket was widened by 2.9 Å (*Figure 5F*). As established before (*Schoof et al., 2021a*), a similar pocket widening is observed in the I-State structure of eIF2B (2.6 Å induced by eIF2α-P binding). This widening is predicted to prevent eIF2 from properly engaging the fourth binding site on eIF2Bδ' and hence turns the decameric eIF2B into conjoined tetramers such that only three of the four eIF2-eIF2B binding interfaces remain readily accessible to eIF2 binding. Our structural observations, therefore, explain the decrease in eIF2 binding and reduction in GEF activity of the βH160D decamer. The remaining portion of slow phase dissociation of eIF2 from βH160D decamers, though, indicates that engagement with all four interfaces, while disfavored, is not impossible as is the case with the pure tetrameric species. By contrast, the inhibitor (eIF2α-P) binding site (*Figure 5G*) was not changed significantly compared to the eIF2B-eIF2α-P complex structure. This observation is consistent with the similar binding affinities measured for eIF2-P towards the βH160D decamer and the WT decamer. We conclude that the βH160D mutation shifts the eIF2B decamer into a conformation closely resembling the I-State.

## eIF2B βH160D mutation leads to stress-independent ISR activation

Given that the eIF2B βH160D mutation biases eIF2B's conformation toward an I-State like conformation, reducing its GEF activity, we predicted that expression of eIF2B βH160D in cells would lead to constitutive ISR activation. To test this notion, we introduced the βH160D mutation into the genome of HEK293FTR cells by editing the endogenous eIF2Bβ gene (*EIF2B2*) (*Figure 6—figure supplement*

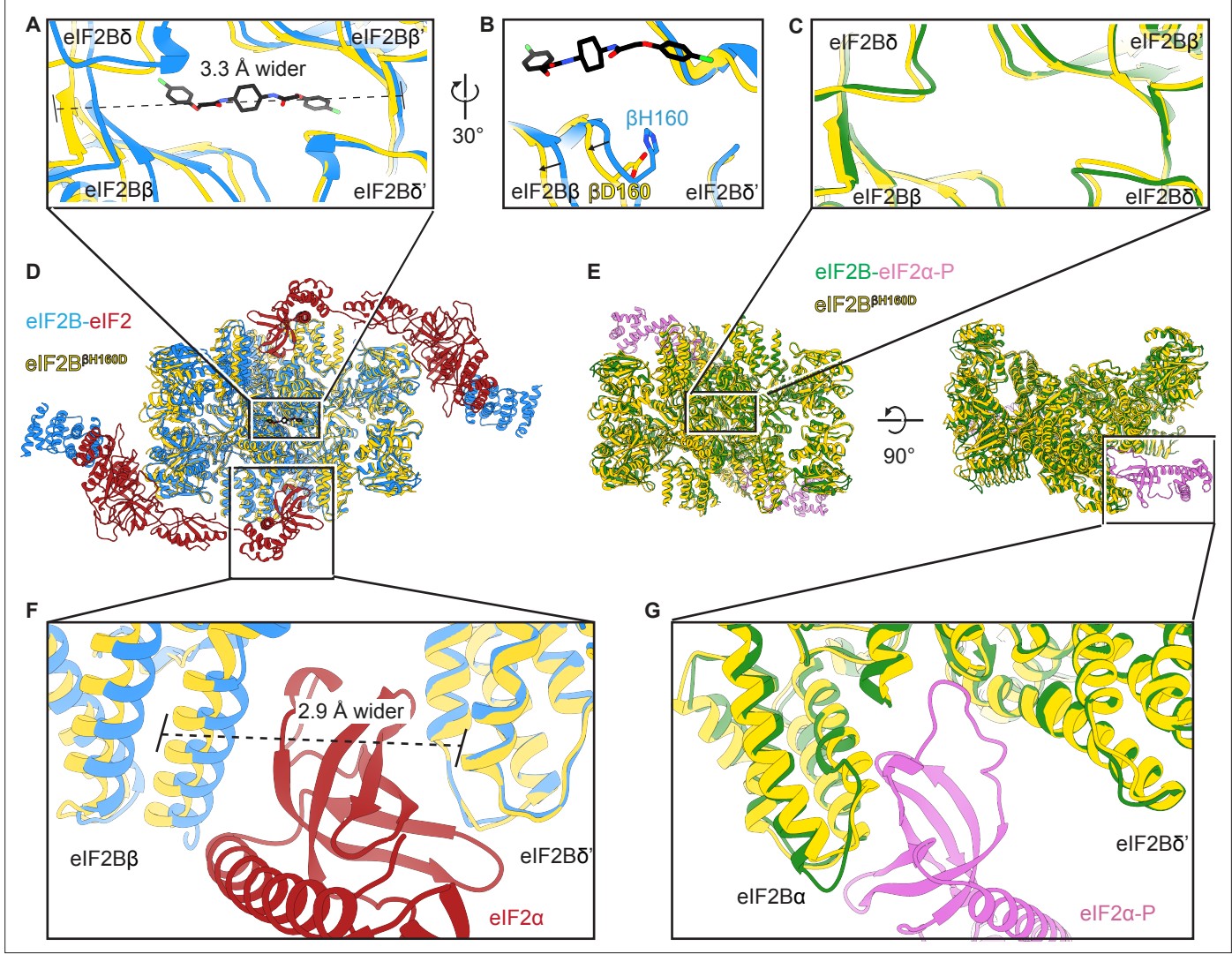

**Figure 5.** The βH160D mutation conformationally diminishes eIF2B activity. (**A**) Overlay of the eIF2B^βH160D structure to the eIF2B-eIF2 structure showing a ~ 3 Å lengthening of the ISRIB-binding pocket in the eIF2B^βH160D structure. The pocket lengthening is measured between eIF2Bδ and eIF2Bδ' L482; the 'prime' indicates the subunit of the opposing tetramer. ISRIB is shown in stick representation. (**B**) A rotated view of panel (**A**) showing that in the eIF2B^βH160D structure the loop bearing βD160 retracts from the opposite tetramer due loss of some attractive interactions (for details, see ***Figure 5—figure supplement 1***). (**C**) Overlay of the eIF2B^βH160D structure to the eIF2B-eIF2α-P structure showing the similar dimensions of the ISRIB binding pockets. (**D**) Zoom out of the overlay in panels (**A**), (**B**), and (**F**). (**E**) Zoom out of the overlay in panel (**C**) and (**G**). (**F**) Overlay of the eIF2-bound eIF2B to eIF2B^βH160D showing the 2.9 Å widening of the eIF2α binding pocket induced by the βH160D mutation. The pocket widening is measured between eIF2Bβ E139 and eIF2Bδ' R250. (**G**) Overlay of the eIF2α-P-bound eIF2B to eIF2B^βH160D showing the similar dimensions of the eIF2α-P binding pockets. Protein molecules are colored as in ***Figure 4***. ISRIB is colored in CPK.

The online version of this article includes the following figure supplement(s) for figure 5:

**Figure supplement 1.** Structural details of the symmetry interface of the WT vs βH160D decamer.

*1A*). Using CRISPR/Cas9 technology, we obtained two such lines. One cell line yielded a homozygous clone in which all alleles were edited (line βH160D #1) (***Figure 6—figure supplement 1B, C***). The other was a heterozygous clone containing one edited allele while the remaining alleles were knocked out through CRISPR/Cas9-induced frameshift mutations (line βH160D #2). Critically, both βH160D cell lines showed eIF2Bβ and eIF2Bε protein levels comparable to the unedited parental cells, demonstrating that the mutation does not destabilize eIF2Bβ or other complex members and that compensatory mechanisms must normalize the gene dosage imbalance in clone #2 (***Figure 6A***; ***Wortham et al., 2016***). We observed constitutive, low-level activation of the ISR in both clones, exemplified by elevated levels of ATF4 protein in the absence of stress (***Figure 6A***, lanes 5 and 9 vs lane

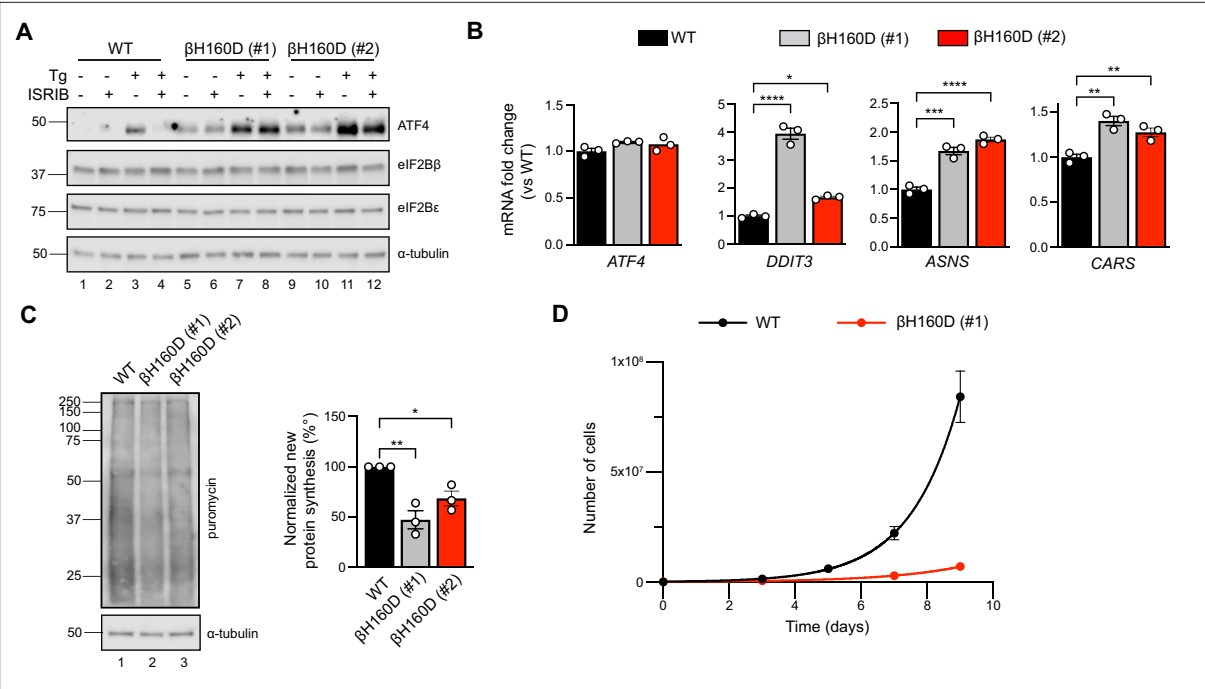

**Figure 6.** The βH160D mutation spontaneously activates the integrated stress response (ISR) in cells. (**A**) Western blot of WT vs *EIF2B2^H160D* HEK293FTR cell lines [βH160D (#1) and βH160D (#2)] treated with and without stress [10 nM thapsigargin (Tg) or ISRIB (200 nM) for 1 hr. eIF2B subunit levels do not differ between cell lines. ATF4 is constitutively produced in the βH160D cell lines (lanes 5 and 9, compare to lane 1), and its induction is ISRIB-insensitive (lanes 6, 8, 10, 12, compare with lane 4). α-tubulin serves as a loading control. (**B**) RT-qPCR for *ATF4* and *ATF4* transcriptional targets in untreated WT vs βH160D cell lines. Transcript levels were normalized to *GAPDH* signal and fold changes were calculated with WT level set to 1. While there is no difference in *ATF4* transcript level, the *ATF4* target genes *DDIT3* (CHOP), *ASNS,* and *CARS* are significantly transcriptionally upregulated in the βH160D lines (one-way ANOVA with Dunnett post-hoc tests). (**C**) Puromycin incorporation assay for new protein synthesis. Left panel: representative blot of cell lysates treated with a 10 min puromycin pulse and blotted for puromycin (new protein synthesis) or tubulin (loading control). Right panel: quantification of puromycin incorporation. The puromycin signal is normalized to tubulin levels and set at 100% for WT. Both βH160D cells show a reduction of basal protein translation [one-way ANOVA with Dunnett post-hoc test, p=0.0026 for WT vs βH160D (#1) and p=0.0288 for WT vs βH160D (#2)]. (**D**) Growth curves showing that βH160D cells grow slower than WT cells WT doubling time = 25.7 hr, s.e.m. = 3.6 hr; βH160D doubling time = 38.4 hr, s.e.m. = 3.5 hr. All error bars and '±' designations are s.e.m. For (**B, D**) n = 3 biological replicates. For (**C**), n = 3 biological replicates, each of which was the average of three technical replicate transfers.*p<0.05, **p<0.01, ***p<0.001, ****p<0.0001.

The online version of this article includes the following source data and figure supplement(s) for figure 6:

**Source data 1.** Raw data for the western blots, qPCR, puromycin-incorporation assay, and cell growth.

**Source data 2.** Original image files for western blots.

**Figure supplement 1.** CRISPR-Cas9 editing of the endogenous *EIF2B2* gene with the βH160D mutation in HEK293FTR cells.

**Figure supplement 2.** Cells with the βH160D mutation in the endogenous *EIF2B2* gene show reduced protein translation.

**Figure supplement 2—source data 1.** Western blots of puromycin incorporation assays.

**Figure supplement 2—source data 2.** Original image files for western blots of puromycin incorporation assays.

**Figure supplement 3.** The *EIF2B2*-H160D mutation does not alter phosphorylated eIF2α levels and is ISRIB resistant.

**Figure supplement 3—source data 1.** Raw data for the western blots and Phostag blots, and cell growth assay.

**Figure supplement 3—source data 2.** Original image files for western blots of eIF2 phosphorylation status.

1). ATF4 induction was still responsive to induced stress with thapsigargin (lanes 7 and 11) but could not be alleviated by ISRIB treatment in the βH160D lines, both in the absence or presence of stressor (*Figure 6A*). ATF4 is translationally upregulated during the ISR and, accordingly, ATF4 mRNA levels remained unchanged between WT and the two βH160D clones (*Figure 6B*). However, as expected, key ATF4 transcriptional targets (such as *DDIT3, ASNS, and CARS*) were upregulated in βH160D cells, confirming that increased ATF4 mRNA translation leads to production of active ATF4, which in turn activates transcription of its downstream stress-responsive genes (*Figure 6B*).

The second hallmark of an active ISR is the general inhibition of translation initiation and, hence, a reduction in protein synthesis. To monitor protein synthesis, we treated WT and βH160D cells with puromycin and assessed puromycin incorporation in nascent polypeptide chains by immunoblotting. Both βH160D cell lines displayed significantly reduced levels of basal protein synthesis (βH160D #1 cells: 47 ± 9.0%; βH160D #2 cells: 69 ± 7.3%, both compared to WT), again consistent with constitutive activation of the ISR (*Figure 6C*, *Figure 6—figure supplement 2*). WT and βH160D cells did not differ in eIF2α phosphorylation levels, underlining the observation that the impairment of eIF2B GEF activity caused by this mutation is sufficient to induce a constitutive ISR (*Figure 6C*, *Figure 6—figure supplement 3A-B*).

Phenotypically, the constitutive ISR activation was accompanied by slow cell growth: cell doubling time increased from 25.7 ± 3.6 h for WT cells to 38.4 ± 3.5 h for βH160D (#1) cells and could not be rescued by ISRIB treatment (*Figure 6D*, *Figure 6—figure supplement 3C*).

## Discussion

Here, we show that a single engineered H to D mutation in eIF2Bβ alters the conformation of the eIF2B decamer, resulting in altered dissociation kinetics of substrate eIF2, a ~ three fold reduction of intrinsic enzymatic activity, and resistance to ISRIB rescue. In cells, this hypomorphic mutation culminates in a constitutively activated low-level ISR. The structural, biochemical, and cellular changes resulting from the βH160D mutation are evocative of the Inhibitor (eIF2-P) bound state of eIF2B ('I-State'). In conjunction with our prior assessment of changes in eIF2B induced by eIF2α-P binding, these orthogonal data underscore how the conformational changes brought about by eIF2α-P binding govern ISR activation (A/I-State model) and that even the presence of eIF2α-P is dispensable as long as an I-State or I-State like conformation is maintained. eIF2B is a far more dynamic complex than we realized just a year ago. Small molecules (ISRIB and its derivatives), the natural substrate (eIF2), and viral proteins (SFSV NSs) can stabilize eIF2B in its active A-State (*Kashiwagi et al., 2021*; *Schoof et al., 2021b*; *Schoof et al., 2021a*; *Zyryanova et al., 2021*). Conversely, binding of the inhibitor (eIF2-P) can compete with these molecules by shifting the decamer to the inhibited I-State (*Schoof et al., 2021a*; *Zyryanova et al., 2021*). Although the conformational displacements induced by βH160D are in many aspects similar to those of the eIF2-P bound I-State when compared to the A-State, they are not identical. While the cryo-EM data show a comparable widening of the eIF2α binding pocket, the movement of the β-solenoid in eIF2Bε is less pronounced in βH160D decamers than in the eIF2-P bound I-State (*Figure 4—figure supplement 2*), likely because the rocking motion induced by βH160D originates near the ISRIB pocket, not from the eIF2-P binding site. In addition, despite extensive classification calculations, we did not recover single-particle images of the βH160D complex belonging to the A-State, arguing against the idea that the βH160D structure is a mixture of A-State and I-State structures. The βH160D decamer rather represents a continuous distribution of conformations with a more restricted range of motion compared to the WT decamer, and for which the average converges to an I-State like model. Hence, acknowledging both similarities and differences to the I-State, we refer to the conformation induced by βH160D as 'I-State like'.

The conformational changes brought about by eIF2-P binding result in a specific enzymatic activity (quantified in the specificity constant $k_{cat}/K_M$) that is approximately 2 orders of magnitude reduced from that of the A-State (*Schoof et al., 2021a*). By comparison, the βH160D mutation causes the specificity constant to drop by only ~2 fold (*Figure 2*). Nevertheless, despite the comparatively small change in eIF2B activity, the mutation induces constitutive ISR activation, suggesting that cells are sensitive to small fluctuations in eIF2B GEF activity. These numbers also tell us that there is still potential for more robust ISR activation. Indeed, treating βH160D cells with relatively low amounts (10 nM) of an eIF2-P inducing stressor like thapsigargin further enhances ATF4 translation (*Figure 6A*). The latter result also suggests that the mutation is compatible with even more potent inhibition mediated by eIF2-P binding. This conclusion is further supported by our 3D reconstructions and the SPR studies, which show that the βH160D mutation does not appreciably affect eIF2-P binding.

We demonstrate that both intrinsic enzymatic activity and substrate (eIF2) binding are affected in the I-State like βH160D decamer. It remains unclear how the conformational changes in either this structure or that in the eIF2-P bound I-State (*Schoof et al., 2021a*) engender a reduced $k_{cat}$, especially given that βH160 is located far from the catalytic center. Non-ideal positioning of substrate molecules that still engage an I-State or I-State like decamer may explain the reduced rate of nucleotide exchange.

Further speculation regarding the mechanism is limited by a lack of structural data for certain critical regions. The eIF2Bε catalytic domain is absent from all but the substrate (eIF2) bound structures. The eIF2Bε linker, a known regulatory region connecting the catalytic domain to the core of eIF2Bε, is similarly unresolved, as are the poorly understood C-terminal solenoid 'ear domains' of eIF2Bγ (*Welsh and Proud, 1993*). The conformation and positioning of these and other regions may be affected during the ISR and play roles in regulation of eIF2B's activity that warrant further examination. With the recent discovery that eIF2B can directly read out and respond to sugar phosphate levels, there may be a host of functions and mechanisms of regulation for eIF2B still to be uncovered (*Hao et al., 2021*).

Our SPR data (*Figure 3*) demonstrate that the effects of the βH160D mutation on substrate (eIF2) binding result from changes to the relative proportion of rapidly dissociating eIF2 molecules. Substrate association, however, remains unaffected. The biphasic dissociation behavior, usually observed for multivalent ligands due to avidity effects, is not entirely unexpected. Substrate-bound structures of eIF2B decamer previously revealed four binding interfaces (IF1–IF4) between eIF2 and eIF2B. Hence, it is possible that stochastic partial binding occurs for a fraction of substrate molecules when the IF4 interface is too distant from IF3 for both to be engaged by eIF2. eIF2α-P binding (or the βH160D mutation) pulls IF4 away from IF3, increasing the probability of this partially engaged binding mode, thus reducing the substrate binding affinity. Notably, though, the biphasic dissociation is not observed for inhibitor (eIF2-P) binding, where both association and dissociation can be fit to monophasic models. This observation suggests greater conformational flexibility along the combinatorial eIF2 binding surfaces than along the eIF2-P binding surfaces.

The βH160 residue is highly conserved amongst eukaryotes. To date, no variation has been reported at this position in the human genome. However, the mechanism by which the βH160D mutation impacts eIF2B activity raises the possibility that certain VWMD mutations may likewise compromise eIF2B function through alteration of conformational state. The disease-associated βE213G mutation (ClinVar VCV000004336), for example, localized near the ISRIB pocket and far away from the catalytic center, reportedly does not affect complex association but substantially reduces GEF activity (*Li et al., 2004*). Understanding the precise mechanism of eIF2B inactivation in individual VWMD patients may be critical for patient stratification and proper treatment. Although ISRIB is unable to rescue the βH160D defect, it is plausible that other analogs (or molecules acting at a different site) with higher affinities than ISRIB may be able to overcome the charge repulsion and restore the A-State conformation, demonstrating the importance of continued endeavors to uncover molecules and strategies to inhibit or activate the ISR orthogonally.

# Materials and methods

Cloning *eIF2B2* (encoding eIF2Bβ) and *eIF2B4* (encoding eIF2Bδ) had previously been inserted into sites 1 and 2 of pACYCDuet-1 and then further edited to include mNeonGreen and a (GGGGS)$_2$ linker at the C-terminus of *eIF2B2* and mScarlet-i and a (GGGGS)$_2$ linker at the C-terminus of *eIF2B4* (pMS029). In-Fusion HD cloning was used to edit this plasmid further and insert the H160D mutation into *eIF2B2* (pMS114).

For CRISPR editing of the *EIF2B2* gene, guide RNAs were designed using the Benchling CRISPR gRNA Design Tool, selecting the guide with the best on-target and off-target scores, and the H160D mutation within 10 bp of the cut site. Cloning of the guide into the guide expression plasmid (MLM3636, with human U6 promoter) was done as previously described (*Kwart et al., 2017*). In brief, the guide RNA sequence was synthesized as single stranded DNA oligos (C005_F and C005_R) that were first annealed at 2 μM in 1× annealing buffer (40 mM Tris-HCl pH 8.0, 20 mM MgCl$_2$, 50 mM NaCl, 1 mM EDTA pH 8.0), for 5 min at 95°C followed by gradual decrease of –0.1°C s$^{-1}$ to 25°C. The MLM3636 plasmid was digested using BsmBI (NEB) in NEB Buffer 3.1 for 2 hr at 55°C, and the 2.2 kb backbone was isolated from a 0.8% agarose gel with 1× SYBR Safe, and purified using the NucleoSpin Gel and PCR cleanup kit (Macherey Nagel). Backbone and annealed guide template were ligated for 1 hr at room temperature using T4 DNA Ligase (NEB), 100 ng backbone, 100 nM guide template, and 1× T4 DNA Ligase buffer (NEB).

## Purification of human eIF2B subcomplexes

Human eIF2Bα$_2$ (pJT075), Avi-tagged eIF2Bα$_2$ (pMS026), WT eIF2Bβγδε (pJT073 and pJT074 co-expression), eIF2Bβ$^{H160D}$γδε (pJT102 and pJT074), Avi-tagged eIF2Bβγδε (pMS001 and pJT074

co-expression), WT eIF2Bβδγε-F tetramers (pMS029 and pJT074 co-expression), and βH160D eIF2Bβδγε-F tetramers (pMS114 and pJT074 co-expression) were purified as previously described (*Tsai et al., 2018*; *Schoof et al., 2021a*).

## Purification of heterotrimeric human eIF2

Human eIF2 was purified as previously described (*Wong et al., 2018*). This material was a generous gift of Calico Life Sciences LLC.

## Analytical ultracentrifugation

Analytical ultracentrifugation (sedimentation velocity) experiments were performed as previously described using the ProteomeLab XL-I system (Beckman Coulter) (*Tsai et al., 2018*). In brief, samples were loaded into cells in a buffer of 20 mM HEPES-KOH, pH 7.5, 150 mM KCl, 1 mM TCEP, and 5 mM $MgCl_2$. A buffer only reference control was also loaded. Samples were then centrifuged in an AN-50 Ti rotor at 40,000 rpm at 20°C and 280 nm absorbance was monitored. Subsequent data analysis was conducted with Sedfit using a non-model-based continuous c(s) distribution.

## In vitro FRET assays

In vitro FRET assays were performed as previously described (*Schoof et al., 2021a*).

## Guanine nucleotide exchange assay

In vitro detection of GDP binding to eIF2 was performed as described previously (*Schoof et al., 2021a*). As before, we first monitored the loading of fluorescent BODIPY-FL-GDP to eIF2. Purified human eIF2 (100 nM) was incubated with 100 nM BODIPY-FL-GDP (Thermo Fisher Scientific) in assay buffer (20 mM HEPES-KOH, pH 7.5, 100 mM KCl, 5 mM $MgCl_2$, 1 mM TCEP, and 1 mg ml $^{-1}$BSA) to a volume of 18 µl in 384 square-well black-walled, clear-bottom polystyrene assay plates (Corning). For the assay buffer, TCEP and BSA were always freshly added the day of the experiment. For the tetramer GEF assays, a 10 X GEF mix was prepared containing 1 µM eIF2Bβγδε tetramer (WT or βH160D), 2% N-methyl-2-pyrrolidone (NMP), and with or without 10 µM ISRIB, again in assay buffer. For the assay, 2 µl of the 10× GEF mix was spiked into the eIF2::BODIPY-FL-GDP mix, bringing the final concentrations to 100 nM tetramer, 0.2% NMP and with or without 1 µM ISRIB. Fluorescence intensity was recorded every 10 s for 40 s prior to the 10× GEF mix spike, and after the spike for 60 min, using a Clariostar PLUS (BMG LabTech) plate reader (excitation wavelength: 477 nm, bandwidth 14 nm; emission wavelength: 525 nm, bandwidth: 30 nm).

For assays with eIF2B decamers (WT or βH160D), decamers were first assembled by combining eIF2Bβγδε tetramer (WT or βH160D) with eIF2Bα$_2$ dimer in a 1:1 molar ratio (a twofold excess of eIF2Bα$_2$ dimer compared to the number of eIF2B(βγδε)$_2$ octamers) at room temperature for at least 30 min. The 10× GEF mix for decamer assays contained 100 nM eIF2B(αβγδε)$_2$ decamer (WT or βH160D) in assay buffer. The ensuing steps were performed as described for the GEF assays with tetramers. Immediately after the loading assay, in the same wells, we spiked in unlabeled GDP to 1 mM to measure unloading, again recording fluorescence intensities every 10 s for 60 min as before. These data were fit to a first-order exponential. For clarity, datapoints were averaged at 1 min intervals and then plotted as single datapoints in *Figure 2*.

## Michaelis–Menten kinetics

The Michaelis-Menten kinetic analysis of eIF2B(αβγδε)$_2$ decamer (WT or βH160D) GEF activity was performed as described previously, with some minor modifications (*Schoof et al., 2021a*). Briefly, BODIPY-FL-GDP loading assays were performed as described above, keeping final decamer concentrations at 10 nM, but varying substrate concentration from 0 to 4 µM. BODIPY-FL-GDP concentration was kept at 2 µM final. The initial velocity was determined by a linear fit to time points acquired at 5 s intervals from 50 to 200 s after addition of decamer. To convert fluorescence intensities to pmol substrate, the gain in signal after 60 min was plotted against eIF2 concentration for the 31.5 nM – 1 µM concentrations. $V_{max}$ and $K_M$ were determined by fitting the initial velocities as a function of eIF2 concentration to the Michaelis–Menten equation in GraphPad Prism 9. For statistical comparisons of $V_{max}$ and $K_M$, we used a two-sided t-test with $\alpha = 0.05$, comparing $V_{max}$ or $K_M$ derived from the individual fit of each replicate experiment.

Affinity determination and variable association analysis by surface plasmon resonance eIF2 and eIF2-P affinity determination experiments were performed on a Biacore T200 instrument (Cytiva Life Sciences) by capturing the biotinylated WT eIF2B decamer, βH160D eIF2B decamer, and WT eIF2B tetramer at ~50 nM on a Biotin CAPture Series S sensor chip (Cytiva Life Sciences) to achieve maximum response (Rmax) of under ~150 response units (RUs) upon eIF2 or eIF2-P binding. eIF2-P was prepared by mixing 5 µM eIF2 in 50-fold excess of 100 nM PERK kinase and with 1 mM ATP. The mixture was incubated at room temperature for 60 min before incubation on ice until dilution into the titration series. 2-fold serial dilutions of purified eIF2 or eIF2-P were flowed over the captured eIF2B complexes at 30 µl min$^{-1}$ for 60 s followed by 600 s of dissociation flow. Following each cycle, the chip surface was regenerated with 3 M guanidine hydrochloride. A running buffer of 20 mM HEPES-KOH, pH 7.5, 100 mM KCl, 5 mM MgCl$_2$, and 1 mM TCEP was used throughout. The resulting sensorgrams were fit in GraphPad Prism 8.0. Association was fit for all species using the association then dissociation model. For eIF2-P binding this model was used to fit dissociation as well. For eIF2 binding, dissociation was fit using the two phase decay model. For eIF2 binding to WT tetramer the data could be modeled with one phase association, one phase dissociation kinetics by setting the percent fast phase dissociation to 100%. For variable association experiments, WT and βH160D eIF2B decamer was immobilized as described above. A solution containing 62.5 nM eIF2 was flowed over the captured eIF2B for 5–480 s at 30 µl min$^{-1}$ to reach the equilibrium of % fast phase dissociation vs % slow phase dissociation. Association was followed by 480 s of dissociation flow. The dissociation phase was then fit in GraphPad Prism 8.0 using the two phase decay model as described above.

## Generation of endogenous βH160D cells

Editing of the *EIF2B2* gene to introduce the H160D mutation in HEK293Flp-In TRex (HEK293FTR) cells was performed using CRISPR-Cas9 according to a previously published protocol, with some minor modifications (*Kwart et al., 2017*). Cells were seeded at 250,000 cells well$^{-1}$ of a 12-well plate and grown for 24 hr prior to transfection with a PAGE-purified, phosphorothioate-protected single-stranded oligonucleotide donor (ssODN) for homologous recombination (C015) (*Renaud et al., 2016*), a plasmid containing Cas9-GFP, and a plasmid encoding the guide RNA (MLM3636-C005). The 100 nt ssODN was designed to simultaneously introduce the H160D missense mutation (CAC to GAC), to add a silent XbaI restriction site at L156 (TCTGGA to TCTAGA), and to block re-digestion by Cas9 after recombination. Transfection was done with Xtreme Gene9 reagent according to the manufacturer's protocol, using a 3:1 ratio of reagent (µl) to DNA (µg). Reagent-only and pCas9-GFP controls were included. Two days post transfection, cells were trypsinized, washed twice in ice-cold filter-sterilized FACS buffer (25 mM HEPES pH 7.0, 2 mM EDTA, 0.5% v/v fetal bovine serum, in 1× PBS), and resuspended in FACS buffer with 400 ng ml$^{-1}$ 7-AAD viability dye (Invitrogen) at around 1 million cells ml$^{-1}$ in filter-capped FACS tubes. Single GFP$^+$, 7-AAD$^-$ cells were sorted into recovery medium (a 1:1 mix of conditioned medium, and fresh medium with 20% fetal bovine serum, 2 mM L-Glutamine, 1 mM sodium pyruvate, and 1× non-essential amino acids) in single wells of 96-well plates using the Sony SH800 cell sorter. The survival rate was around 2% after 2–3 weeks. Surviving clones were expanded and first screened for correct editing by PCR and XbaI restriction digest. For this, genomic DNA was isolated using the PureLink Genomic DNA mini kit (Invitrogen), and a 473 bp fragment of the *EIF2B2* gene was amplified by PCR using 300 nM forward and reverse primers (C001_F and C001_R), 300 µM dNTPs, 1× HF buffer, 100 ng genomic DNA 100 µl$^{-1}$ reaction and 2 U 100 µl$^{-1}$ reaction of KAPA HiFi polymerase for 3 min at 95°C; and 30 cycles of 98°C for 20 s, 68.9°C for 15 s, 72°C for 15 s, prior to cooling at 4°C. PCR reactions were purified using NucleoSpin Gel and PCR cleanup kit (Macherey Nagel), and HighPrep PCR Cleanup beads (MagBio Genomics) using the manufacturer's instructions. Cleaned up products were digested using XbaI restriction enzyme (NEB) in 1× CutSmart buffer and run on a 1.5% agarose gel with 1× SYBR Safe (Invitrogen) and 100 bp DNA ladder (Promega). Clones with an XbaI restriction site were then deep sequenced to confirm correct editing and zygosity. For this, the *EIF2B2* gene was amplified by PCR using 300 nM forward and reverse primers (C034_F and C034_R), 300 µM dNTPs, 1× HF buffer, 100 ng genomic DNA 100 µl$^{-1}$ reaction and 2 U 100 µl$^{-1}$ reaction of KAPA HiFi polymerase for 3 min at 95°C; and 30 cycles of 98°C for 20 s, 64.9°C for 15 s, 72°C for 15 s, prior to cooling at 4°C. The 196 bp product was purified from a 1.5% agarose gel with 1× SYBR Safe using NucleoSpin Gel and PCR cleanup kit (Macherey Nagel), and HighPrep PCR Cleanup beads (MagBio Genomics) using the manufacturer's instructions. A subsequent second PCR added

the Illumina P5/P7 sequences and barcode for deep sequencing. For this, we used 15 ng purified PCR product per 100 µl reaction, 300 nM forward and reverse primer (C036_F_bcx, and C036_R), and 1× KAPA HiFi HotStart mix, for 3 min at 95°C, and 8 cycles of 20 s at 98°C, 15 s at 63.7°C, and 15 s at 72°C prior to cooling on ice. PCR reactions were purified using HighPrep beads (MagBio Genomics), and amplicon quality and size distribution was checked by chip electrophoresis (BioAnalyzer High Sensitivity kit, Agilent). Samples were then sequenced on an Illumina MiSeq (150 bp paired-end), and results were analyzed with CRISPResso (*Pinello et al., 2016*). All cell lines were negative for mycoplasma contamination. Amplicon sequencing data was deposited in NCBI's Sequence Read Archive (SRA) under accession number PRJNA821864.

## Growth curves

Cells were seeded at 100,000 cells well$^{-1}$ of a six-well plate and grown at 37°C and 5% $CO_2$. At confluency, cells were trypsinized, expanded into larger plates, and counted. This was repeated until the WT cells reached confluency in a T225 flask. For drug treatment conditions (*Figure 6—figure supplement 3C*), we used 500 nM ISRIB with DMSO at a final concentration of 0.1% across conditions.

## Western blotting

Cells were seeded at 400,000 cells well$^{-1}$ of a six-well plate and grown at 37°C and 5% $CO_2$ for 24 hr. For drug treatment, we used 10 nM thapsigargin (Tg) (Invitrogen) and 200 nM ISRIB (made in-house) for 1 hr, ensuring the final DMSO concentration was 0.1% across all conditions. For the protein synthesis assay, puromycin was added to a final concentration of 10 µg ml$^{-1}$ for 10 min. Plates were put on ice, cells were washed once with ice-cold phosphate-buffered saline (PBS), and then lysed in 150 µl ice-cold lysis buffer (50 mM Tris-HCl pH 7.4, 150 mM NaCl, 1 mM EDTA, 1% v/v Triton X-100, 10% v/v glycerol, 1× cOmplete protease inhibitor cocktail [Roche], and 1× PhosSTOP [Roche]). Cells were scraped off, collected in an eppendorf tube, and put on a rotator for 30 min at 4°C. Debris was pelleted at 12,000 *g* for 20 min at 4°C, and supernatant was removed to a new tube on ice. Protein concentration was measured using the bicinchonic acid (BCA) assay. Within an experiment, total protein concentration was normalized to the least concentrated sample (typically all values were within ~ 10%). A 5× Laemmli loading buffer (250 mM Tris-HCl pH 6.8, 30% glycerol, 0.25% bromophenol blue, 10% SDS, 5% β-mercaptoethanol) was added to each sample to 1×, and samples were denatured at 95°C for 12 min, then cooled on ice. Wells of AnyKd Mini-PROTEAN TGX precast protein gels (AnyKD, Bio-Rad) were loaded with equal amounts of total protein (around 10 µg), in between Precision Plus Dual Color protein ladder (BioRad). After electrophoresis, proteins were transferred onto a nitrocellulose membrane at 4°C, and then blocked for 2 hr at room temperature in PBS with 0.1% Tween (PBS-T) + 3% milk (blocking buffer) while rocking. Primary antibody staining was performed with gentle agitation at 4°C overnight using the conditions outlined in *Table 3*. After washing four times in blocking buffer, secondary antibody staining was performed for 1 hr at room temperature using anti-rabbit HRP or anti-mouse HRP secondary antibodies (Promega, 1:10,000) in blocking buffer. Membranes were washed 3× in blocking buffer and then 1× in PBS-T without milk. Chemiluminescent detection was performed using SuperSignal West Dura or Femto HRP substrate (Thermo Fisher Scientific), and membranes were imaged on a LI-COR Odyssey gel imager for 0.5–10 min depending on band intensity.

**Table 3.** Western blot antibodies and conditions.

| Antibody target | Host | Dilution | Manufacturer | Cat. number | Blocking conditions |
|---|---|---|---|---|---|
| eIF2Bβ | Rabbit | 1/1000 | ProteinTech | 11034–1-AP | PBS-T + 3% milk |
| eIF2Bε | Mouse | 1/1000 | Santa Cruz Biotechnology | sc-55558 | PBS-T + 3% milk |
| ATF4 | Rabbit | 1/1000 | Cell Signaling | 11,815 S | PBS-T + 3% milk |
| α-tubulin | Mouse | 1/1000 | Cell Signaling | 3873T | PBS-T + 3% milk |
| Puromycin | Mouse | 1/10,000 | Millipore | MABE343 | PBS-T + 3% milk |
| eIF2α | rabbit | 1/1000 | Cell Signaling | 5324 S | PBS-T + 3% milk |
| eIF2α-P (S51) | rabbit | 1/1000 | Cell Signaling | 9721 S | PBS-T + 1% BSA |

**Table 4.** Primers and oligos.

| Oligo | Sequence | Target gene |
|---|---|---|
| B002_F | TGCACCACCAACTGCTTAGC | *GAPDH* |
| B002_R | GGCATGGACTGTGGTCATGAG | *GAPDH* |
| D006_F | ATGACCGAAATGAGCTTCCTG | *ATF4* |
| D006_R | GCTGGAGAACCCATGAGGT | *ATF4* |
| D007_F | GGAAACAGAGTGGTCATTCCC | *DDIT3 (CHOP)* |
| D007_R | CTGCTTGAGCCGTTCATTCTC | *DDIT3 (CHOP)* |
| D070_F | GGAAGACAGCCCCGATTTACT | *ASNS* |
| D070_R | AGCACGAACTGTTGTAATGTCA | *ASNS* |
| D073_F | CCATGCAGACTCCACCTTTAC | *CARS* |
| D073_R | GCAATACCACGTCACCTTTTTC | *CARS* |
| C001_F | ACTTTAAGCACATTAACCCTG | *EIF2B2* |
| C001_R | ACTTGATCTTCTCAGTGTCTC | *EIF2B2* |
| C015 | t*G*CAAAACCGTTCTTACAGAAG GGACAATGGAGAACATTGCAGCCCA GGCTCTAGAGCACATTGACTCCAATGA GGTGATCATGACCATTGGCTTCTCCCGAACAGT | NA (ssODN) |
| C034_F | CGCGTAATGTGTGTTTGTGA | |
| C034_R | GCCTCTACTGTTCGGGAGAA | |
| C036_F_bcx | CAAGCAGAAGACGGCATACGAGATxxxxxx GTGACTGGAGTTCAGACGTGTGCTCTTCCG ATCTCGCGTAATGTGTGTTTGTGA | |
| C036_R | AATGATACGGCGACCACCGAGATCTAC ACTCTTTCCCTACACGA | |
| C005_F | acaccgGGAGCACATTCACTCCAATGg | |
| C005_R | aaaacCATTGGAGTGAATGTGCTCCcg | |

*phosphorothioate bond.
x = barcode nucleotide, different for each clone.

For the phospho-retention blots, equal amounts of total protein lysates (around 10 µg) were loaded on 12.5% Supersep Phos-tag gels (Wako Chemicals) in between Wide-view III protein ladder (Wako Chemicals). After electrophoresis, the gel was washed 3× in transfer buffer with 10 mM EDTA prior to transfer onto nitrocellulose. Blocking, antibody staining and detection was performed as described above.

## RT-qPCR

Cells were seeded at 400,000 cells well$^{-1}$ of a 12-well plate and grown at 37°C and 5% $CO_2$ for 24 hr. The day of RNA extraction, medium was removed, and cells were lysed in 350 µl TriZOL reagent (Invitrogen). All further handling was done in a fume hood decontaminated for the presence of RNAses using RNAse ZAP (Invitrogen). Total RNA was isolated using the DirectZOL RNA miniprep kit (Zymo Research), including an on-column DNase digest, according to the manufacturer's instructions. RNA concentration was measured using Nanodrop. cDNA was synthesized using 600 ng input total RNA per 40 µl reaction with the iScript cDNA Synthesis Kit (BioRad), cycling for 5 min at 25°C, 20 min at 46°C, and 1 min at 95°C. Samples were cooled and diluted 1/5 in Rnase-free water. qPCR reactions were set up with final 1× iQ SYBR Green supermix (BioRad), 400 nM each of Fw and Rev QPCR primers (see *Table 4*), 1/5 of the diluted cDNA reaction, and RNAse-free water. No-template and no-reverse transcription reactions were included as controls. Reactions were run in triplicates as 10 µl reactions in 384-well plates on a BioRad CFX384 Thermocycler, for 3 min at 95°C, and then 40 cycles of 95°C for 10 s and 60°C for 30 s, ending with heating from 55 to 95°C in 0.5°C increments for melting curve generation. $C_q$s and melting curves were calculated by the BioRad software. $C_q$ values of technical replicates were averaged, and values were calculated with the ΔΔCt method using *GAPDH* for reference gene normalization. Graph points reflect fold changes compared to WT vehicle, with bars being

the mean ± s.e.m. Statistical analysis was done using GraphPad Prism 9 on log-transformed values with ordinary one-way ANOVA and Dunnett's post-hoc test.

## Sample preparation for cryo-electron microscopy

Decameric eIF2Bβ$^{H160D}$ was prepared by incubating 16 μM eIF2Bβ$^{H160D}$γδε with 8.32 μM eIF2Bα$_2$ in a final solution containing 20 mM HEPES-KOH, 200 mM KCl, 5 mM MgCl$_2$, and 1 mM TCEP. This 8 μM eIF2B(αβ$^{H160D}$γδε)$_2$ sample was further diluted to 750 nM. For grid freezing, a 3 μl aliquot of the sample was applied onto the Quantifoil R 1.2/1/3 400 mesh Gold grid and we waited for 30 s. A 0.5 μl aliquot of 0.1–0.2% Nonidet P-40 substitute was added immediately before blotting. The entire blotting procedure was performed using Vitrobot (FEI) at 10°C and 100% humidity.

## Electron microscopy data collection

Cryo-EM data was collected on a Titan Krios transmission electron microscope operating at 300 keV. Micrographs were acquired using a Gatan K3 direct electron detector. The total dose was 67 e$^-$/ Å$^2$, and 117 frames were recorded during a 5.9 s exposure. Data was collected at 105,000× nominal magnification (0.835 Å/pixel at the specimen level), with a nominal defocus range of –0.6 to –2.0 μm.

## Image processing

The micrograph frames were aligned using MotionCor2 (*Zheng et al., 2017*). The contrast transfer function (CTF) parameters were estimated with GCTF (*Zhang, 2016*). For the decameric eIF2Bβ$^{H160D}$, Particles were picked in Cryosparc v2.15 using the apo eIF2B (EMDB: 23209) as a template (*Punjani et al., 2017*; *Schoof et al., 2021a*). Particles were extracted using an 80-pixel box size and classified in 2D. Classes that showed clear protein features were selected and extracted for ab initio reconstruction, followed by homogenous refinement. Particles belonging to the best class were then re-extracted with a pixel size of 2.09 Å, and then subjected to homogeneous refinement, yielding a reconstruction of 4.25 Å. These particles were subjected to another round of heterogeneous refinement followed by homogeneous refinement to generate a consensus reconstruction consisting of the best particles. These particles were re-extracted at a pixel size of 0.835 Å. Then, CTF refinement was performed to correct for the per-particle CTF as well as beam tilt. A final round of nonuniform refinement yielded the final structure of 2.8 Å.

## Atomic model building, refinement, and visualization

For the decameric eIF2Bβ$^{H160D}$, the previously published apo eIF2B model (PDB ID: 7L70) was used as a starting model (*Schoof et al., 2021a*). Each subunit was docked into the EM density individually and then subjected to rigid body refinement in Phenix (*Adams et al., 2010*). The models were then manually adjusted in Coot and then refined in phenix.real_space_refine using global minimization, secondary structure restraints, Ramachandran restraints, and local grid search (*Emsley and Cotwan, 2004*). Then iterative cycles of manual rebuilding in Coot and phenix.real_space_refine were performed. The final model statistics were tabulated using Molprobity (*Chen et al., 2010*). Distances were calculated from the atomic models using UCSF Chimera (*Pettersen et al., 2004*). Molecular graphics and analyses were performed with the UCSF Chimera package. UCSF Chimera is developed by the Resource for Biocomputing, Visualization, and Informatics and is supported by NIGMS P41-GM103311. The atomic model is deposited into PDB under the accession code 7TRJ. The EM map is deposited into EMDB under the accession code EMD-26098.

## Acknowledgements

We thank the Walter lab for helpful discussions throughout this project; Geeta Narlikar for input on enzyme kinetics; Calico for a generous gift of purified eIF2 heterotrimer; and Z Yu and D Bulkley of the UCSF Center for Advanced Cryo-EM facility, which is supported by NIH grants S10OD021741 and S10OD020054 and the Howard Hughes Medical Institute (HHMI); We also thank the QB3 shared cluster for computational support and Hehua and Quixote for inspiration and emotional support. Funding This work was supported by generous support from Calico Life Sciences LLC (to PW); a generous gift from The George and Judy Marcus Family Foundation (To PW); the Belgian-American Educational Foundation (BAEF) Postdoctoral Fellowship (to MB), the Damon Runyon Cancer Research Foundation Postdoctoral fellowship (to LW); the Jane Coffin Child Foundation Postdoctoral Fellowship

(to RL); a Chan Zuckerberg Biohub Investigator award and an HHMI Faculty Scholar grant (AF). PW is an Investigator of the Howard Hughes Medical Institute.

## Additional information

### Competing interests

Peter Walter: is an inventor on U.S. Patent 9708247 held by the Regents of the University of California that describes ISRIB and its analogs. Rights to the invention have been licensed by UCSF to Calico. For the remaining authors, no competing financial interests exist. The other authors declare that no competing interests exist.

### Funding

| Funder | Grant reference number | Author |
|---|---|---|
| Calico Life Sciences LLC | | Peter Walter |
| The George and Judy Marcus Family Foundation | | Peter Walter |
| Belgian-American Educational Foundation | | Morgane Boone |
| Jane Coffin Child Foundation | | Rosalie E Lawrence |
| Chan Zuckerberg Biohub Investigator Award | | Adam Frost |
| Howard Hughes Medical Institute | Faculty Scholar grant | Adam Frost |
| Howard Hughes Medical Institute | | Peter Walter |
| Damon Runyon Cancer Research Foundation | | Lan Wang |

The funders had no role in study design, data collection and interpretation, or the decision to submit the work for publication.

### Author contributions

Morgane Boone, Lan Wang, Michael Schoof, Conceptualization, Formal analysis, Investigation, Visualization, Writing - original draft, Writing – review and editing; Rosalie E Lawrence, Investigation, Writing – review and editing; Adam Frost, Methodology, Supervision, Writing – review and editing; Peter Walter, Funding acquisition, Supervision, Writing – review and editing

### Author ORCIDs

Morgane Boone ![ORCID] http://orcid.org/0000-0002-7807-5542
Lan Wang ![ORCID] http://orcid.org/0000-0002-8931-7201
Rosalie E Lawrence ![ORCID] http://orcid.org/0000-0003-3386-7391
Adam Frost ![ORCID] http://orcid.org/0000-0003-2231-2577
Peter Walter ![ORCID] http://orcid.org/0000-0002-6849-708X
Michael Schoof ![ORCID] http://orcid.org/0000-0003-3531-5232

### Decision letter and Author response

Decision letter https://doi.org/10.7554/eLife.76171.sa1
Author response https://doi.org/10.7554/eLife.76171.sa2

## Additional files

### Supplementary files
• Transparent reporting form

## Data availability

All data generated or anaysed during this study are included in the manuscript and source data files. The final structural model has been deposited in PDB under the accession code 7TRJ. Amplicon sequencing data for the CRISPR clones has been deposited in NCBI's Sequence Read Archive (SRA) under accession number PRJNA821864.

The following datasets were generated:

| Author(s) | Year | Dataset title | Dataset URL | Database and Identifier |
|---|---|---|---|---|
| Schoof M | 2022 | A point mutation in the nucleotide exchange factor eIF2B constitutively activates the integrated stress response by allosteric modulation | https://www.ncbi.nlm. nih.gov/bioproject/? term=PRJNA821864 | NCBI BioProject, PRJNA821864 |
| Schoof M | 2022 | A point mutation in the nucleotide exchange factor eIF2B constitutively activates the integrated stress response by allosteric modulation | https://www.rcsb.org/ structure/7TRJ | RCSB Protein Data Bank, 7TRJ |

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
