## [Editor Report]

The paper describes the consequences of a missense mutation in the β subunit of the eIF2B complex that advances the understanding of the mechanisms of action of eIF2B in controlling the integrated stress response. The combination of biochemical, structural, and in-cell experiments constitutes a comprehensive study that supports a model for allosteric regulation of the active/inactive states of the eIF2B complex. The findings are relevant to neuropathologies, infectious and inflammatory diseases, diabetes, and metabolic disorders.

---

## [Decision Letter]

**Decision letter after peer review:**

Thank you for submitting your article "A point mutation in the nucleotide exchange factor eIF2B constitutively activates the integrated stress response by allosteric modulation" for consideration by *eLife*. Your article has been reviewed by 2 peer reviewers, and the evaluation has been overseen by a Reviewing Editor and Suzanne Pfeffer as the Senior Editor. The reviewers have opted to remain anonymous.

Essential revisions:

The paper is highly appropriate for for publication in *eLife* but it would benefit from some experimental additions. Reviewer #1 suggested (see below) for Figure 6 to add S51-P western, ATF4-luc reporter analysis, test ISRIB impact on mutant cell growth. Reviewer #2 made similar comments. Some additional analyses (no new experiments) to address the structural questions are raised by reviewer #2 (#3-5). The authors need to mention the related gcd- and gcn- mutations in yeast eIF2B (from Alan Hinnebusch's published work) – especially the gcn- mutation gcd7-D178Y located only three residues away from the H160D mutation studied in this paper.

*Reviewer #1 (Recommendations for the authors):*

1. Line 43: "initiator methionine tRNA" – "tRNA" is missing.

2. Line 131: text is confusing; perhaps "of" could be inserted after "regulation".

3. Does Tg lead to hyperactivation of ATF4 expression in the H180D cells? An ATF4 reporter assay would likely be the simplest means to examine this.

4. Line 493: Panel "(C)" should be "(D)".

5. While perhaps not required for this paper, analysis of published yeast eIF2B mutations might be insightful when considering the A/I model of eIF2B. The partial loss-of-function phenotype of the eIF2beta-H160D mutation is reminiscent of the Gcd- phenotypes described by Alan Hinnebusch for numerous mutations in all five subunits of yeast eIF5B. While the yeast eIF2B complex appears to lack the ISRIB binding site, I wonder if examination of the characterized eIF2B mutations would provide further support for the A/I state model proposed for eIF2B. Of particular note, the Hinnebusch lab has characterized different sets of mutations in the eIF2B subunits that either lead to loss of eIF2B activity (Gcd- or GCN3-constitutive) or insensitivity to inhibition by phosphorylated eIF2 (Gcn-). Intriguingly, I saw that a gcd7-D178Y mutation in yeast eIF2Bbeta that is only three residues before the H160 residue leads to a Gcn- phenotype (see Vazquez de Aldana, MCB 14:3208-3222, 1994 and Pavitt, MCB 17:1298-1313, 1998). Could this mutation uncouple the allostery between the regulatory and active sites in eIF2B? It is also noteworthy that some yeast eIF2Balpha (GCN3-constitutive) mutations can be suppressed by mutations in eIF2alpha (SUI2), perhaps providing in vivo data to support the structural models of eIF2alpha binding to the regulatory site.

*Reviewer #2 (Recommendations for the authors):*

1. Figure 1: It would be helpful to show the quality of the purified eIF2B factors (tetramer and dimer) as a supplemental figure. Include the statistical analyses for panels E and F.

2. Figure 6: Should consider using the ATF4-luciferase reporter in the mix of measurements for constitutive partial activation of the ISR by the H160D β subunit mutant. This would reinforce the idea that ISR induction of ATF4 is predominantly by preferential translation and explain why the ATF4 mRNA was not measurably changed in basal conditions between the WT and H160D cells.

3. The mechanistic basis proposed for a change in conformation driven by charge repulsion between D160 in the β subunit and D450 in the δ subunit seems improbable based on the conformations observed in the inhibited state (6O9Z). Here the closest approach of His 160 to Asp 450 is 5 Å. This distance is also 5 Å in the activated state 6081. Given that modeling of D160 in this position would be even farther away at closest approach (at least 5.5 Å) and that H160 is about 3.2 Å from E183 within the β subunit, it is not clear how the authors have come to this model for charge repulsion. At the very least, the manuscript should provide a figure that shows β subunit H160 in the wild type structure and D450 in the δ subunit superimposed with D160 and D450 in the structure reported here.

4. The authors should provide the RCSB ID for the deposited structure and the validation information ensuring that the structure meets expected standards.

5. The manuscript states in the discussion that despite extensive classifications, they did not find any particles that correspond to the A-state of eIF2B. Did the authors use the A-state for template picking? Would one be able to distinguish the I-state from the A-state at the low resolution normally used for 2D classifications? The manuscript havs provided images only for 2D classifications from template picking.

At the very least, the manuscript should provide images obtained for 2D and 3D classifications without template picking and provide an estimate of what percentage of the particles correspond to the state assigned in the final structure. One explanation for an intermediate hinge angle of 3.5 degrees is that you have an average of two states, one corresponding to the 6 degree state found in the inhibited state relative to the activated state and the other corresponding to that found in the activated state.

---

## [Author Response]

Reviewer #1 (Recommendations for the authors):1. Line 43: "initiator methionine tRNA" – "tRNA" is missing.

Corrected.

2. Line 131: text is confusing; perhaps "of" could be inserted after "regulation".

Corrected.

3. Does Tg lead to hyperactivation of ATF4 expression in the H180D cells? An ATF4 reporter assay would likely be the simplest means to examine this.

Yes, as can be observed on the westerns blots (Figure 6A, lanes 7/8 and 11/12) and as we mention in the Discussion (third paragraph), Tg treatment further increases ATF4 signal compared to vehicle-treated mutant cells. The H160D mutation is thus hypomorphic and there is room for more inhibition; phosphorylation of eIF2 through activation of the PERK kinase (as occurs with Tg treatment) further inhibits eIF2B-H160D, leading to an increase in ATF4 translation compared to the vehicle-treated cells. This is in line with our in vitro Michaelis-Menten data showing a 2-3-fold decrease in specific enzyme activity with the H160D mutation (Figure 2C and D), which is orders of magnitude less than the ~100-fold decrease caused by addition of phosphorylated eIF2 (Schoof et al., *eLife*, 2021). As we argued above, a direct inquiry of ATF4 protein levels readily answers this question.

4. Line 493: Panel "(C)" should be "(D)"

Corrected.

5. While perhaps not required for this paper, analysis of published yeast eIF2B mutations might be insightful when considering the A/I model of eIF2B. The partial loss-of-function phenotype of the eIF2beta-H160D mutation is reminiscent of the Gcd- phenotypes described by Alan Hinnebusch for numerous mutations in all five subunits of yeast eIF5B. While the yeast eIF2B complex appears to lack the ISRIB binding site, I wonder if examination of the characterized eIF2B mutations would provide further support for the A/I state model proposed for eIF2B. Of particular note, the Hinnebusch lab has characterized different sets of mutations in the eIF2B subunits that either lead to loss of eIF2B activity (Gcd- or GCN3-constitutive) or insensitivity to inhibition by phosphorylated eIF2 (Gcn-). Intriguingly, I saw that a gcd7-D178Y mutation in yeast eIF2Bbeta that is only three residues before the H160 residue leads to a Gcn- phenotype (see Vazquez de Aldana, MCB 14:3208-3222, 1994 and Pavitt, MCB 17:1298-1313, 1998). Could this mutation uncouple the allostery between the regulatory and active sites in eIF2B? It is also noteworthy that some yeast eIF2Balpha (GCN3-constitutive) mutations can be suppressed by mutations in eIF2alpha (SUI2), perhaps providing in vivo data to support the structural models of eIF2alpha binding to the regulatory site.

Indeed, the reviewer makes a good point that, should the A/I-State model of regulation hold true in yeast, I-State stabilization could be an explanation for the Gcd phenotype of certain yeast eIF2B mutations, similarly to what we have observed for the bH160D mutation in human eIF2B. Conversely, it is possible that other mutations may stabilize the complex in the A-State, which would lead to a Gcn phenotype in yeast. The gcd7-D178Y mutation could be an example of the latter as it makes yeast eIF2B insensitive to inhibition by eIF2-P (so the opposite effect of the H160D mutation). No allosteric chain could reliably be traced from the available human structures, making it tricky to predict the effect of any mutation on the coupling of regulatory and active sites without in vitro evidence.

It is interesting to note, however, that the allosteric changes observed and validated in mammalian eIF2B remain to be confirmed in yeast. Curiously, overlaying published structures of *S. pombe* eIF2B (apo, eIF2alpha bound or eIF2alpha-P bound) do not show the same rocking motion that we and others observe when aligning human eIF2B A vs I-State. There are other discrepancies between human and yeast eIF2B, such as cryo-EM and X-ray structures showing the binding of substrate eIF2 in the same orientation as inhibitory eIF2-P. The a subunit of eIF2B is also not essential in yeast whereas it is essential in humans and presumably other mammals. These data could point to evolutionary divergence in regulatory mechanisms, or yet-to-pinpoint differences in experimental preparation. It is therefore difficult to draw any conclusions about human eIF2B A/I state regulation from the available yeast data, and indeed, a thorough reexamination of yeast eIF2B regulation may be in order.

Reviewer #2 (Recommendations for the authors):1. Figure 1: It would be helpful to show the quality of the purified eIF2B factors (tetramer and dimer) as a supplemental figure. Include the statistical analyses for panels E and F.

we now included a Coomassie-stained SDS-PAGE gel image of the purified protein complexes used in this study as a supplemental figure (Figure 1 —figure supplement 1). For panels E and F, the error on the datapoints is indicated by error bars representing the standard error of mean, as denoted in the figure legend. We have also updated the figure legend to include the median and s.e.m. for the EC_50_ of assembly by ISRIB and eIF2Ba_2_.

2. Figure 6: Should consider using the ATF4-luciferase reporter in the mix of measurements for constitutive partial activation of the ISR by the H160D β subunit mutant. This would reinforce the idea that ISR induction of ATF4 is predominantly by preferential translation and explain why the ATF4 mRNA was not measurably changed in basal conditions between the WT and H160D cells.

See our above response to reviewer #1 regarding the addition of an ATF4 reporter. We do not consider it appropriate or necessary to introduce an ATF4-luciferase reporter in different cell lines and compare the resulting lines side-by-side. Our western blots show that ATF4 protein levels, but not transcript levels, are up in two independent mutants compared with WT, which is indicative of translational regulation.

3. The mechanistic basis proposed for a change in conformation driven by charge repulsion between D160 in the β subunit and D450 in the δ subunit seems improbable based on the conformations observed in the inhibited state (6O9Z). Here the closest approach of His 160 to Asp 450 is 5 Å. This distance is also 5 Å in the activated state 6081. Given that modeling of D160 in this position would be even farther away at closest approach (at least 5.5 Å) and that H160 is about 3.2 Å from E183 within the β subunit, it is not clear how the authors have come to this model for charge repulsion. At the very least, the manuscript should provide a figure that shows β subunit H160 in the wild type structure and D450 in the δ subunit superimposed with D160 and D450 in the structure reported here.

The reviewer is correct in that the single repulsion between the two aspartates is probably not the only cause of the structural changes. We note that the H160D mutation also causes the loss of a stacking interaction with a phenylalanine on the opposite side, and additionally repositions R228 to locally reach a new stable state. We have updated the main text and Figure 5, and added a new supplementary figure (Figure 5 —figure supplement 1) to reflect the local structural rearrangments caused by the H160D mutation. Please refer to the main text and figure for details.

4. The authors should provide the RCSB ID for the deposited structure and the validation information ensuring that the structure meets expected standards.

We now included the PDB ID and EMDB ID in the manuscript.

5. The manuscript states in the discussion that despite extensive classifications, they did not find any particles that correspond to the A-state of eIF2B. Did the authors use the A-state for template picking? Would one be able to distinguish the I-state from the A-state at the low resolution normally used for 2D classifications? The manuscript havs provided images only for 2D classifications from template picking.At the very least, the manuscript should provide images obtained for 2D and 3D classifications without template picking and provide an estimate of what percentage of the particles correspond to the state assigned in the final structure. One explanation for an intermediate hinge angle of 3.5 degrees is that you have an average of two states, one corresponding to the 6 degree state found in the inhibited state relative to the activated state and the other corresponding to that found in the activated state.

We do not expect our picking method to bias against A-state particles, as the templates we used for particle picking are generated from the apo eIF2B structure, which assumes an A-state conformation. Regarding the reviewer’s comment on whether the H160D dataset could be an average of two states, we performed an in-depth analysis of the cryo-EM data used in this work, compared it to the data that generated the apo eIF2B structure, and reached a model for how the mutation changes the conformational distribution of the eIF2B decamers. In short, our analysis shows that the H160D structure most likely represents a continuous distribution of different conformations, rather than a mix of two discrete populations. For details, please see the new supplemental figure we made that illustrate this analysis (Figure 4 —figure supplement 3).